# Balancing Privacy and Performance for Private Federated Learning Algorithms

## Abstract

Federated learning (FL) is a distributed machine learning (ML) framework where multiple clients collaborate to train a model without exposing their private data. FL involves cycles of local computations and bi-directional communications between the clients and server. To bolster data security during this process, FL algorithms frequently employ a differential privacy (DP) mechanism that introduces noise into each client's model updates before sharing. However, while enhancing privacy, the DP mechanism often hampers convergence performance. In this paper, we posit that an optimal balance exists between the number of local steps and communication rounds, one that maximizes the convergence performance within a given privacy budget. Specifically, we prove the optimal number of local steps and communication rounds that enhance the convergence bounds of the DP version of the ScaffNew algorithm. Our findings reveal a direct correlation between the optimal number of local steps, communication rounds, and a set of variables, e.g the DP privacy budget and other problem parameters, specifically in the context of strongly convex optimization. We furthermore provide empirical evidence to validate our theoretical findings.

## 1 Introduction

Recent success of machine learning (ML) can be attributed to the increasing size of both ML models and their training data without significantly modifying existing well-performing architectures. This phenomenon has been demonstrated in several studies, e.g., Sun et al. (2017); Kaplan et al. (2020); Chowdhery et al. (2022); Taylor et al. (2022). However, this approach is infeasible since it needs to store a massive training dataset in a single location.

**Federated learning.** To address this issue, federated learning (FL) Konečný et al. (2016); Konečný et al. (2016b;a) has emerged as a distributed framework, where many clients collaborate to train ML models by sharing only their local updates while keeping their local data for security and privacy concerns Dwork et al. (2014); Apple (2017); Burki (2019); Viorescu et al. (2017). Two types of FL include (1) *cross-device FL* which leverages millions of edge, mobile devices, and (2) *cross-silo FL* where clients are data centers or companies and the client number is very small. Both FL types pose distinct challenges and are suited for specific use cases Kairouz et al. (2021). While cross-device FL solves problems over the network of statistically heterogeneous clients with low network bandwidth in IoT applications Nguyen et al. (2021), cross-silo FL is characterized by high inter-client dataset heterogeneity in healthcare and bank domains Kairouz et al. (2021); Wang et al. (2021). In this paper, we focus mainly on cross-silo FL algorithms which are usually efficient and scalable due to low communication costs among a very few clients at each step.

**Differential privacy.** Although private data is only kept at each client in FL, clients' local updates can still leak a lot of information about their private data, Shokri et al. (2017); Zhu et al. (2019). This necessitates several tools for ensuring privacy for FL. Privacy-preserving variations of FL algorithms therefore have been proposed in the literature, based on the concept of differential privacy (DP) Dwork et al. (2014) to bound the amount of information leakage. To provide privacy guarantees of FL algorithms [1], we can apply

---

[1] For the detailed review of the DP training methods, we recommend (Ponomareva et al., 2023, Section 4).

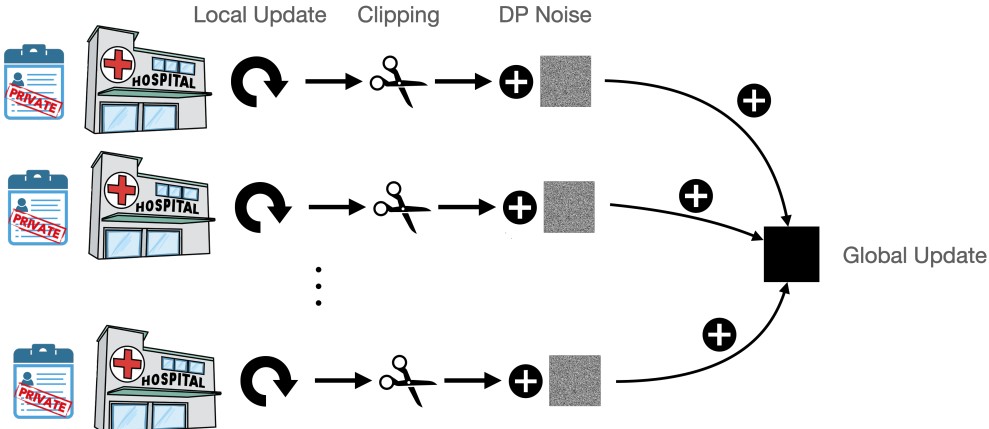

Figure 1: Visualization of differentially private federated methods: Each client computes its local update, which is then clipped and partially masked using DP noise. The adjusted update is then aggregated and used by all clients to update the global model.

DP mechanisms at clients, Terrail et al. (2022); Truex et al. (2020); Sun et al. (2020); Kim et al. (2021); Geyer et al. (2017); Abadi et al. (2016). These client-level DP mechanisms enhance privacy by clipping and then injecting noise into clients' local updates before they are communicated in each communication round of running DP federated algorithms (see Figure 1). These mechanisms prevent attackers from deducing original data even though they obtain perturbed gradients.

**Privacy-and-utility trade-off.** While enhancing privacy, the DP mechanisms exacerbates convergence performance of DP federated algorithms. This motivates the study of a set of hyper-parameters for DP federated algorithms that optimally balance privacy and convergence speed. For instance, Wei et al. (2020) proves that DP-FedProx algorithms have the optimal number of communication rounds that guarantee the highest convergence performance given a privacy budget. Nonetheless, their algorithm requires solving the proximal updates exactly on each local client, and their convergence and utility are guaranteed under very restrictive assumptions. In particular, the optimal number of communication rounds exists only for the cases when (1) the client number and privacy level are high enough, and (2) the Euclidean distance between the client's local gradient and the global gradient is sufficiently low (i.e., each client has the same unique minimizer).

**Contributions.** The goal of this paper is to show the optimal number of local steps and communication rounds for DP federated learning algorithms for a given privacy budget. Our contributions are summarized as follows:

- We analyze the DP version of the ScaffNew algorithm under standard but non-restrictive assumptions. Our analysis reveals there is an optimal number of local steps and communication rounds each client should take for solving strongly convex problems. Unlike Wei et al. (2020), we provide an explicit expression for the optimal number of total communication rounds that achieves the best model performance at a fixed privacy budget.

- We verify our theory in empirical evaluations showing that the optimal number of local steps and communication rounds exist for DP-FedAvg and DP-ScaffNew. In particular, these DP algorithms with optimally tuned parameters can achieve almost the same convergence performance as their non-private algorithms.

**Notations.** For $x, y \in \mathbb{R}^d$, $\langle x, y \rangle := x^\top y$ is the inner product and $\|x\| = \sqrt{\langle x, x \rangle}$ is the $\ell_2$-norm. A continuously differentiable function $f : \mathbb{R}^d \to \mathbb{R}$ is $\mu$-strongly convex if there exists a positive constant $\mu$ such

that for $x, y \in \mathbb{R}^d$

$$f(y) \geq f(x) + \langle \nabla f(x), y - x \rangle + \frac{\mu}{2} \|y - x\|^2,$$

and has $L$-Lipschitz continuous gradient if for all $x, y \in \mathbb{R}^d$

$$\|\nabla f(y) - \nabla f(x)\| \leq L\|y - x\|.$$

Finally, $\Pr(C)$ is the probability of event $C$ happening.

## 1.1 Related Work

Now, we review existing literature closely related to our work in federated learning and differential privacy.

**Federated learning.** Two classical algorithms in federated learning include (1) FedAvg, which updates its ML model by averaging local stochastic gradient updates McMahan et al. (2017), and (2) FedProx, which computes its ML model by aggregating local proximal updates Li et al. (2020); Yuan & Li (2022). The convergence of both algorithms has been extensively studied under the data heterogeneity assumption. These classical algorithms suffer from slow convergence due to the small step-size range resulting from the high level of data heterogeneity among the clients. Several other federated algorithms have been developed to enhance the training performance of FedAvg and FedProx. For example, Proxskip Mishchenko et al. (2022), SCAFFOLD Karimireddy et al. (2020), FedSplit Pathak & Wainwright (2020) and FedPD Zhang et al. (2021) leverage proximal updates, variance reduction, operator-splitting schemes, and ADMM techniques, respectively.

**Differential privacy.** Differential privacy (DP) Dwork et al. (2014) is a standard technique for characterizing the amount of information leakage. A fundamental mechanism to design DP algorithms is the Gaussian mechanism Dwork et al. (2014), which adds the Gaussian noise to the output before it is released. The variance of a DP noise is adjusted according to the sensitivity function, which is upper-bounded by the clipping threshold and the Lipschitz continuity of objective functions. The DP guarantee of running DP algorithms for $K$ steps can be obtained by the (advanced) composition theorem Dwork et al. (2014). Recent tools such as Rényi Differential Privacy Mironov (2017) and the moments accountant Abadi et al. (2016) allow to obtain tighter privacy bounds for the Gaussian mechanism under composition. In the context of FL, many works attempted to develop DP federated learning algorithms with strong client-level privacy and utility guarantees, e.g., DP-FedAvg Zhao et al. (2020); McMahan et al. (2017), DP-FedProx Wei et al. (2020), and DP-SCAFFOLD Noble et al. (2022).

## 2 DP Federated Learning Algorithm

To show the optimal number of local steps and communication rounds for DP federated algorithms, we consider the following federated minimization under privacy constraints:

$$\underset{x \in \mathbb{R}^d}{\text{minimize}} \left[ f(x) := \frac{1}{N} \sum_{i=1}^{N} f_i(x) \right], \tag{1}$$

where $N$ is the number of clients, $f_i(x)$ is the loss function of $i^{\text{th}}$ client based on its own local data, and $x \in \mathbb{R}^d$ is a vector storing global model parameters.

**Local differential privacy.** To quantify information leakage, we use local differential privacy (local DP) Dwork et al. (2014). Local DP relies on the notion of neighboring sets, where we say that two federated datasets $D$ and $D'$ are neighbors if they differ in only one client. Local DP aims to protect the privacy of each client whose data is being used for learning the ML model by ensuring that the obtained model does not reveal any sensitive information about them. A formal definition follows.

---

**Algorithm 1** DP-FedAvg

---

1: **Input:** Initial point $x_0 \in \mathbb{R}^d$, the number of communication rounds $T \geq 1$, the number of local steps $\tau \geq 1$, step-size $\eta > 0$, clipping threshold $C > 0$, DP noise variance $\sigma^2$
2: **for** $t = 0, 1, \ldots, T - 1$ **do**
3:      **for** each client $i \in \{1, 2, \ldots, N\}$ in parallel **do**
4:          Set $x_{t,0}^{(i)} = x_t$
5:          **for** $j = 0, \ldots, \tau - 1$ **do**
6:              Compute stochastic gradient $g_i(x_{t,j}^{(i)})$
7:              Update $x_{t,j+1}^{(i)} = x_{t,j}^{(i)} - \eta g_i(x_{t,j}^{(i)})$
8:          **end for**
9:          Compute $\text{clip}(x_{t,\tau}^{(i)} - x_t)$ where $\text{clip}(x) := \min\left(1, \frac{C}{\|x\|}\right) x$
10:         Send $\Delta_t^{(i)} = \text{clip}(x_{t,\tau}^{(i)} - x_t) + \mathcal{N}(0, \sigma^2 I)$
11:      **end for**
12:      Global averaging: $x_{t+1} = x_t + \frac{1}{N} \sum_{i=1}^{N} \Delta_t^{(i)}$.
13: **end for**
14: **return** $x_T$

---

**Definition 1** (Dwork et al. (2014)). A randomized algorithm $\mathcal{A} : \mathcal{D} \rightarrow \mathcal{O}$ with domain $\mathcal{D}$ and range $\mathcal{O}$ is $(\epsilon, \delta)$-differentially private if for all neighboring federated datasets $D, D' \in \mathcal{D}$ and for all events $S \subset \mathcal{O}$ in the output space of $\mathcal{A}$, we have

$$\Pr(\mathcal{A}(D) \in S) \leq e^\epsilon \cdot \Pr(\mathcal{A}(D') \in S) + \delta.$$

**DP-FedAvg.** DP-FedAvg McMahan et al. (2017) is the DP version of popular FedAvg for solving equation 1 with formal privacy guarantees. In each communication round $t = 0, \ldots, T - 1$, all the $N$ clients in parallel update the global model parameters $x_t$ based on their local progress with the DP mask $\Delta_t^{(i)}$. Here, all $\Delta_t^{(i)}$ are communicated by the all-to-all communication primitive and are defined by:

$$\Delta_t^{(i)} = \text{clip}(x_{t,\tau}^{(i)} - x_t) + \mathcal{N}(0, \sigma^2 I),$$

where $x_{t,\tau}^{(i)}$ is the local model parameter of client $i$ from running $\tau$ stochastic gradient descent steps based on their local data and the current global model parameters $x_t$. This DP-masked local progress $\Delta_t^{(i)}$ guarantees local DP by two following steps Abadi et al. (2016); Dwork et al. (2014): (1) all clients clip their local progress $x_{t,\tau}^{(i)} - x_t$ with the clipping threshold $C > 0$ which bounds the influence of each client on the global update, and (2) each clipped progress is perturbed by independent Gaussian noise with zero mean and variance $\sigma^2$ that depends on the DP parameters $\epsilon, \delta$ and the number of communication rounds $T$. We provide the visualization of this DP-masking procedure in Fig. 1, and the full description of DP-FedAvg in Algorithm 1. However, since FedAvg reaches incorrect stationary points Pathak & Wainwright (2020), we rather consider the DP version of ScaffNew Mishchenko et al. (2022) that eliminates this issue by adding an extra drift/shift to the local gradient.

**DP-ScaffNew.** To this end, we consider DP-ScaffNew to prove that its optimal choices for local steps and communication rounds exist. DP-ScaffNew is the DP version of ScaffNew algorithms Mishchenko et al. (2022), and its pseudocode is in Algorithm 2. Notice that DP-ScaffNew differs from DP-FedAvg in two places. First, each client in DP-ScaffNew adds the extra correction term $h_t^{(i)}$ (line 5, 8 and 13, Alg. 2) to remove the client drift caused by local stochastic gradient descent steps. Second, the number of local steps for DP-ScaffNew is stochastic (line 6, Alg. 2).

To facilitate our analysis, we consider DP-ScaffNew for strongly convex optimization in equation 1. We assume (A) that each local step is based on the full local gradient, i.e., $g_i(x_t^{(i)}) = \nabla f_i(x_t^{(i)})$, and (B) that the clipping operator is never active, i.e., the norm of the update is always less than the clipping value $C$. Assumption (A) is not essential in learning overparameterized models such as deep neural networks,

---

**Algorithm 2** DP-ScaffNew

---

1: **Input:** Initial points $x_0 = x_0^{(1)} = \ldots = x_0^{(N)} \in \mathbb{R}^d$, initial control variates $h_0^{(1)}, \ldots, h_0^{(N)} \in \mathbb{R}^d$ such that $\sum_{i=1}^{N} h_0^{(i)} = 0$, number of iterations $T \geq 1$, probability $p \in (0, 1]$, step-size $\eta > 0$, clipping threshold $C > 0$, DP noise variance $\sigma^2$
2: **for** $t = 0, 1, \ldots, T-1$ **do**
3:     Flip a coin $\theta_t \in \{0, 1\}$ where $\text{Prob}(\theta_t = 1) = p$
4:     **for** each client $i \in \{1, 2, \ldots, N\}$ in parallel **do**
5:        Compute stochastic gradient $g_i(x_t^{(i)})$
6:        Update $\hat{x}_{t+1}^{(i)} = x_t^{(i)} - \eta[g_i(x_t^{(i)}) - h_t^{(i)}]$
7:        **if** $\theta_t = 1$ **then**
8:           Send $\Delta_t^{(i)} = \text{clip}(\hat{x}_{t+1}^{(i)} - x_t) + \mathcal{N}(0, \sigma^2 I)$, where $\text{clip}(x) := \min\left(1, \frac{C}{\|x\|}\right) x$
9:           Global averaging: $x_{t+1} = x_{t+1}^{(i)} = x_t + \frac{1}{N} \sum_{j=1}^{N} \Delta_t^{(j)}$
10:          Compute $h_{t+1}^{(i)} = h_t^{(i)} + \frac{p}{\eta}\left(\frac{1}{N}\sum_{j=1}^{N} \Delta_t^{(j)} - \Delta_t^{(i)}\right)$
11:        **else**
12:           Skip Communication: $x_{t+1}^{(i)} = \hat{x}_{t+1}^{(i)}$, $x_{t+1} = x_t$, $h_{t+1}^{(i)} = h_t^{(i)}$
13:        **end if**
14:     **end for**
15: **end for**

---

consistent linear systems, or classification on linearly separable data. For these models, the local stochastic gradient converges towards zero at the optimal solution Vaswani et al. (2019), i.e. $g_i(x^\star) = 0$. Assumption (B) is crucial as the clipping operator introduces non-linearity into the updates, thus complicating the analysis. However, we show that as the algorithm converges, the norms of the updates decrease, and clipping is only active for the first few rounds. Thus, running the algorithm with or without clipping has minimal effect on the convergence which can refer to Observation 3 in our experimental evaluation section. Further note that the results for DP-ScaffNew also apply for DP-FedAvg to learn the overparameterized model. For this model, each $h_t^{(i)}$ converges towards a zero vector, and thus DP-ScaffNew becomes DP-FedAvg.

Now, we present privacy and utility (convergence with respect to a given local $(\epsilon, \delta)$ DP noise) guarantees for DP-ScaffNew in Algorithm 2 for strongly convex problems. All the derivations are deferred to the appendix.

**Lemma 1** (Local differential privacy for Algorithm 2, Theorem 1 Abadi et al. (2016))**.** There exist constants $u, v \in \mathbb{R}^+$ so that given the expected number of communication rounds $pT$, Algorithm 2 is $(\epsilon, \delta)$-differentially private for any $\delta > 0$, $\epsilon \leq upT$ if

$$\sigma^2 \geq v \frac{C^2 pT \ln(1/\delta)}{\epsilon^2}.$$

Using Lemma 1, we obtain the utility guarantee (convergence under the fixed local $(\epsilon, \delta)$-DP budget) for Algorithm 2.

**Theorem 1** (Utility for Algorithm 2)**.** Consider the optimization problem in equation 1, where each $f_i(x)$ is $\mu$-strongly convex and $L$-smooth. Then, the output of Algorithm 2 with $0 < \eta \leq 1/L$, $0 < p \leq 1$, $g_i(x) = \nabla f_i(x)$, and $C > \|\hat{x}_{t+1}^{(i)} - x_t\|$ for $i \in \{1, \ldots, N\}$ and $t \geq 0$ satisfies $(\epsilon, \delta)$-differentially private and the following: for $T \geq 1$,

$$\mathbf{E}[\psi_T] \leq \theta^T \psi_0 + \frac{p^2 N d}{1 - \theta} \cdot \frac{vC^2 T \ln(1/\delta)}{\epsilon^2}, \tag{2}$$

where $\theta := \max(1 - \mu\eta, 1 - p^2)$, $\psi_t := \|\mathbf{x}_t - \mathbf{x}^\star\|^2 + \frac{\eta^2}{p^2}\|\mathbf{h}_t - \mathbf{h}^\star\|^2$, $\mathbf{x}_t := \left[(x_t^1)^\top, \ldots, (x_t^N)^\top\right]^\top$, $\mathbf{x}^\star := \left[(x^\star)^\top, \ldots, (x^\star)^\top\right]^\top$, $\mathbf{h}_t := \left[(h_t^1)^\top, \ldots, (h_t^N)^\top\right]^\top$, and $\mathbf{h}^\star := \left[(\nabla f_1(x^\star))^\top, \ldots, (\nabla f_N(x^\star))^\top\right]^\top$.

Theorem 1 establishes a linear convergence of Algorithm 2 under standard assumptions on objective functions in equation 1, i.e., the $\mu$-strong convexity and $L$-smoothness of $f_i(x)$. The utility bound in equation 2 consists

of two terms. The first term implies the convergence rate which depends on the learning rate $\eta$, the strong convexity parameter $\mu$, and the algorithmic parameters $p, T$. The second term is the residual error due to the local $(\epsilon, \delta)-$DP noise variance. This error can be decreased by lowering $p, T$ at the price of worsening the optimization term (the first term). To balance the first and second terms, Algorithm 2 requires careful tuning of the learning rate $\eta$, the probability $p$, and the iterations $T$.

**Optimal values of $\eta, p, T$ for DP-ScaffNew.** From equation 2, the fastest convergence rate in the first term can be obtained by setting the largest step-size $\eta^\star = 1/L$ and $p^\star = \sqrt{\mu/L}$, Mishchenko et al. (2022). Given $\eta^\star$ and $p^\star$, we can find $T^\star$ by minimizing the convergence bound in equation 2 by solving:

$$\frac{d}{dT}(\theta^T \psi_0) + \frac{p^2 N d}{1 - \theta} \cdot \frac{v C^2 \ln(1/\delta)}{\epsilon^2} = 0.$$

We hence obtain $\eta^\star, p^\star$, and $T^\star$ that minimize the upper-bound in equation 2 in the next corollary.

**Corollary 1.** Consider Algorithm 2 under the same setting as Theorem 1. Choosing $\eta^\star = 1/L$, $p^\star = \sqrt{\mu/L}$, and

$$T^\star = \frac{\ln\left(\frac{\psi_0 \epsilon^2 \ln([1-\mu/L]^{-1})}{v C^2 N d \ln(1/\delta)}\right)}{\ln\left([1 - \mu/L]^{-1}\right)}$$

minimizes the upper-bound for $\mathbf{E}[\psi_T]$ in equation 2.

To the best of our knowledge, the only result showing the optimal value of local steps and communication rounds that balance privacy and convergence performance of DP federated algorithms is Wei et al. (2020). However, our result is stronger than Wei et al. (2020) as we do not impose the data heterogeneity assumption, the sufficiently large values of the client number $N$, and the privacy protection level $\epsilon$. Our result also provides the explicit expression for optimal hyper-parameters for DP-ScaffNew $\eta^\star, p^\star, T^\star$. Furthermore, from Corollary 1, we obtain the optimal expected number of local steps and of communication rounds, which are $1/p^\star$ and $p^\star T^\star$, respectively.

## 3 Experimental Evaluation

Finally, we empirically demonstrate that the optimal local steps and communication rounds exist for DP federated algorithms, which achieve the balance between privacy and convergence performance. We show this by evaluating DP-FedAvg and DP-ScaffNew, including their non-private versions, for solving various learning tasks over five publicly available federated datasets. In particular, we benchmark DP-FedAvg and DP-ScaffNew for learning neural network models to solve (A) multiclass classification tasks over CIFAR-10 Krizhevsky et al. (2009) and FEMNIST Caldas et al. (2018), (B) binary classification tasks over Fed-IXI Terrail et al. (2022) and Messidor Decencière et al. (2014), and (C) next word prediction tasks over Reddit Caldas et al. (2018). The summary of datasets with their associated learning tasks and hyper-parameter settings is fully described in Table 1. Furthermore, we implemented DP federated algorithms for solving learning tasks over these datasets in PyTorch 2.0.1Paszke et al. (2019) and CUDA 11.8, and ran all the experiments on the computing server with an NVIDIA A100 Tensor Core GPU (40 GB). We shared all source codes for running DP federated algorithms in our experiments as supplementary materials. These source codes will be made available later upon the acceptance of this paper.

**Datasets.** CIFAR-10, subset of FEMNIST, Fed-IXI, and Messidor consist of, respectively, 60000 $32 \times 32$ images with 10 objects, 40263 $128 \times 128$ images with 62 classes (10 digits, 26 lowercase, 26 uppercase), 566 T1-weighted brain MR images with binary classes (brain tissue or no), and 3220 $224 \times 224$ eye fundus images with binary labels (diabetic retinopathy or no). On the other hand, Reddit comprises 56,587,343 comments on Reddit in December 2017. While we directly used CIFAR-10 for training the NN model, the rest of the datasets was pre-processed before the training. All pre-processing details for each data set are in the appendix. Also, we split the CIFAR-10, FEMNIST, and Reddit datasets equally at random among 5, 6, and 3 clients, respectively, while the raw Fed-IXI and Messidor datasets are split by 3 users (which represent hospitals) by default.

|          | Clients | Train S.       | Test S.      | B. Size | Iters | Clip       | Task                    |
|----------|---------|----------------|--------------|---------|-------|------------|-------------------------|
| CIFAR10  | 5       | $10000 \pm 0$  | $2000 \pm 0$ | 64      | 10K   | 10,50,100  | Classification (10)     |
| FEMNIST  | 6       | $6129 \pm 1915$| $684 \pm 213$| 16      | 10K   | 10,50,100  | Classification (63)     |
| Reddit   | 3       | $28750 \pm 0$  | $11807 \pm 0$| 64      | 10K   | 10,50,100  | Language Model          |
| Fed-IXI  | 3       | $151 \pm 95$   | $38 \pm 24$  | 1       | 400   | 10,20,50   | Brain Mask Segmeation   |
| Messidor | 3       | $300 \pm 0$    | $100 \pm 0$  | 4       | 500   | 10,20,50   | Classification (2)      |

Table 1: Summary of datasets used in the experiments with indication of client sample variability ($\pm$ standard deviation).

**Training.** We used a two-layer convolutional neural network (CNN) for the multiclass classification over CIFAR10 and FEMNIST. For the brain mask segmentation and binary classification, we employed a 3D-Unet model over Fed-IXI and a VGG-11 model over Messidor. The 3D-Unet model has the same model parameter tunings and baseline as that in Terrail et al. (2022), but we use group normalization instead of batch normalization to prevent the leakage of data statistics. Finally, a 2-layer long short-term memory (LTSM) network with an embedding and hidden size of 256 is employed to predict the next token in a sequence with a maximum length of 10 tokens from the Reddit data, which is tokenized according to LEAF Caldas et al. (2018). Moreover, the initial weights of these NN models were randomly generated by default in PyTorch.

**Hyper-parameter tunings.** We used SGD for every client in the local update steps for DP-FedAvg and DP-ScaffNew. The learning rate for the local update is fixed at 0.05 for the Reddit dataset and at 0.01 for the rest of datasets. The number of local steps is selected from the set of the all divisors of total iterations for running the algorithms. Thus, the number of communication rounds is always equal to the quotients of the iteration and local step. The total iterations for each dataset are detailed in Table. 1. For every dataset, we test 4 distinct privacy levels represented by $(\epsilon, \delta)$ values of $(3.3, 2, 1, 0.5)$ paired with $10^{-5}$, along with 3 different clip thresholds. These parameter settings are consistent with those used in DP-FedAvg and DP-ScaffNew. Furthermore, Table. 1 provides the train and test sizes for each client, as well as the batch size during training.

**Evaluation and performance metrics.** We collected the results from each experiment from 3 trials and reported the average and standard deviation of metrics to evaluate the performance of algorithms.

We measure the following metrics for each experiment. While we collect *accuracy* as our evaluation metric for classification and next-word prediction tasks, we use *Dice coefficient* to evaluate the performance for segmentation tasks. Given that the value of these metrics falls within the range $[0, 1]$ and a higher value signifies better performance, we define the *test error rate* as

$$test\ error\ rate = 1 - metric,$$

where *metric* can be accuracy or dice coefficient.

Moreover, to analyze the correlation within our data, we resort to the $R^2$ test. In essence, $R^2$ is a statistical measure representing the percentage of the data's variance that our model accounts for. The $R^2$ values at 0 and 1 imply, respectively, no and perfect explanatory power of the model.

### 3.1 Results

We now discuss the results of DP-FedAvg and DP-ScaffNew over benchmark datasets under different $(\epsilon, \delta)$-DP noise and clipping thresholds. We provide the following observations.

**Observation 1.** *The non-trivial optimal number of local steps exist for both DP-FedAvg and DP-ScaffNew.*

We measure the test error rate of DP-FedAvg and DP-ScaffNew with respect to the number of local steps, given the fixed total iteration number and other hyper-parameters. Figure 2 shows that there is an optimal

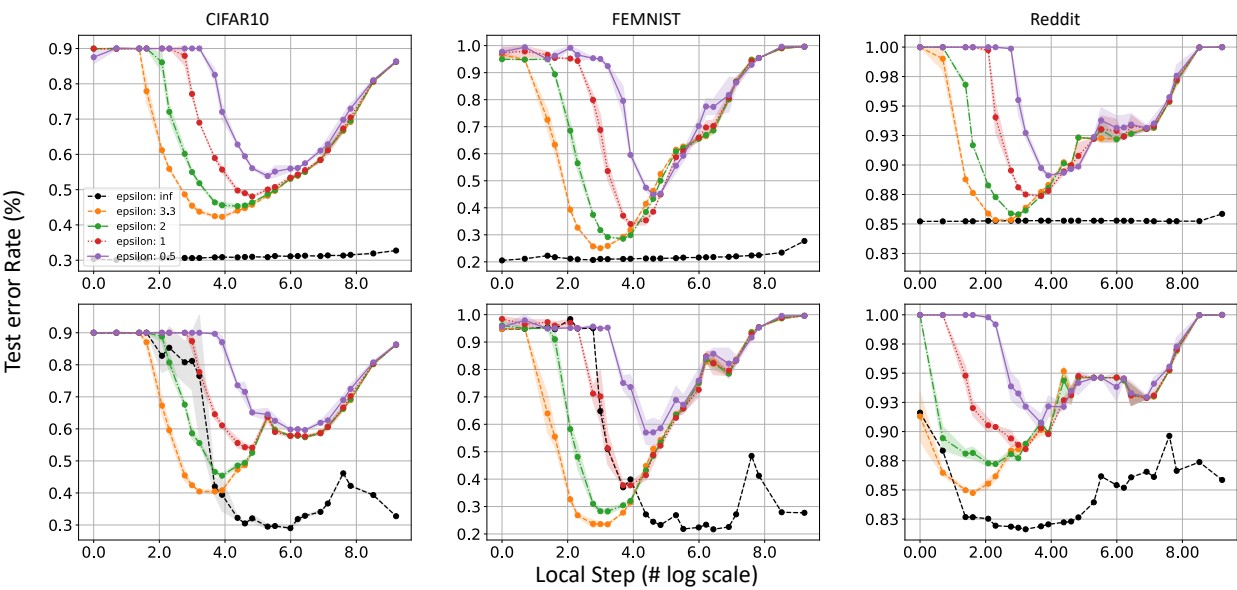

Figure 2: First line: DP-FedAvg, Second line: DP-ScaffNew. Clip threshold: 10.

number of local steps that enabless DP federated algorithms to achieve the lowest test error rate. These DP federated algorithms with optimally tuned local steps achieve performance almost comparable to their non-private algorithms, especially for tasks over most benchmarked datasets (i.e., FEMNIST and Reddit). Also, notice that the optimal local step exists even for DP-ScaffNew without the DP noise (when $\epsilon \to +\infty$). Our results align with theoretical findings for the non-DP version of DP-ScaffNew Mishchenko et al. (2022), and also with Corollary 1 (which implies that $1/p^\star$ represents the optimal number of expected local steps).

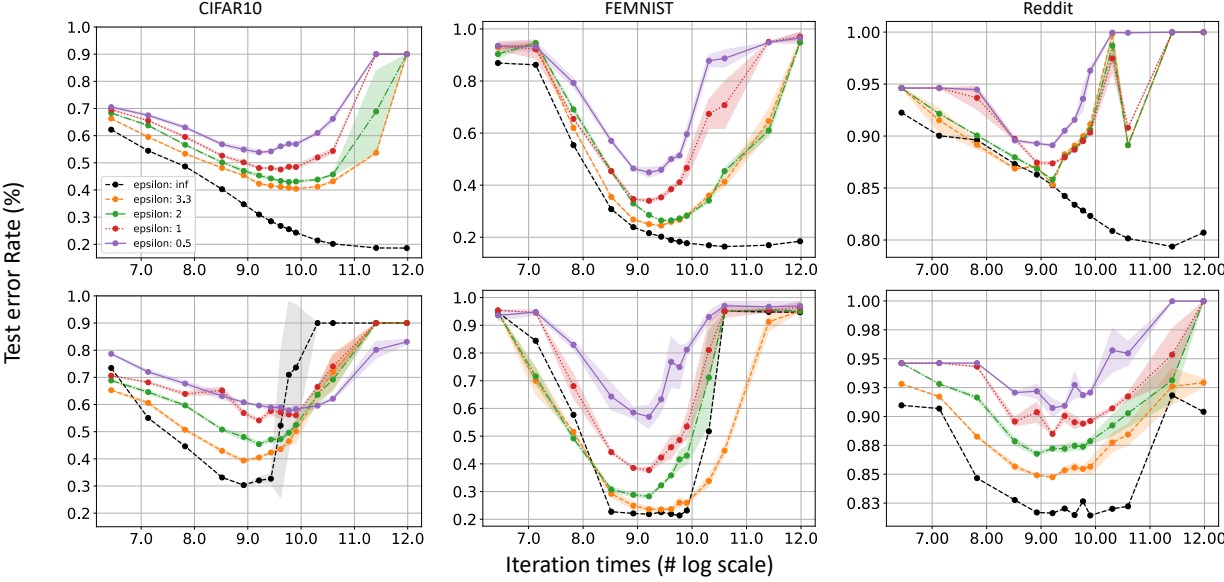

Figure 3: First line: DP-FedAvg, Second line: DP-ScaffNew. Clip threshold: 10. All local step fix to the optimal of its privacy budget.

**Observation 2.** *There exists a non-trivial optimal number of iterations for DP-FedAvg and DP-ScaffNew.*

Figure 3 shows the optimal number of total iterations $T^\star$ exists for DP federated algorithms to achieve the lowest test error rate, thus validating our findings of Corollary 1. We note that as the total iteration number $T$ grows, DP-ScaffNew attains poor performance on both the test and train dataset even in the absence of noise and the clipping operator (black in Figure 3). This phenomenon is not present in DP-FedAvg. We hypothesize that the issue with DP-ScaffNew arises due to its variance reduction approach, which employs SVRG-like control variates. Although this type of variance reduction has demonstrated remarkable theoretical and practical success, it may falter when applied to the hard non-convex optimization problems frequently encountered during the training of modern deep neural networks, as observed by Defazio & Bottou (2019).

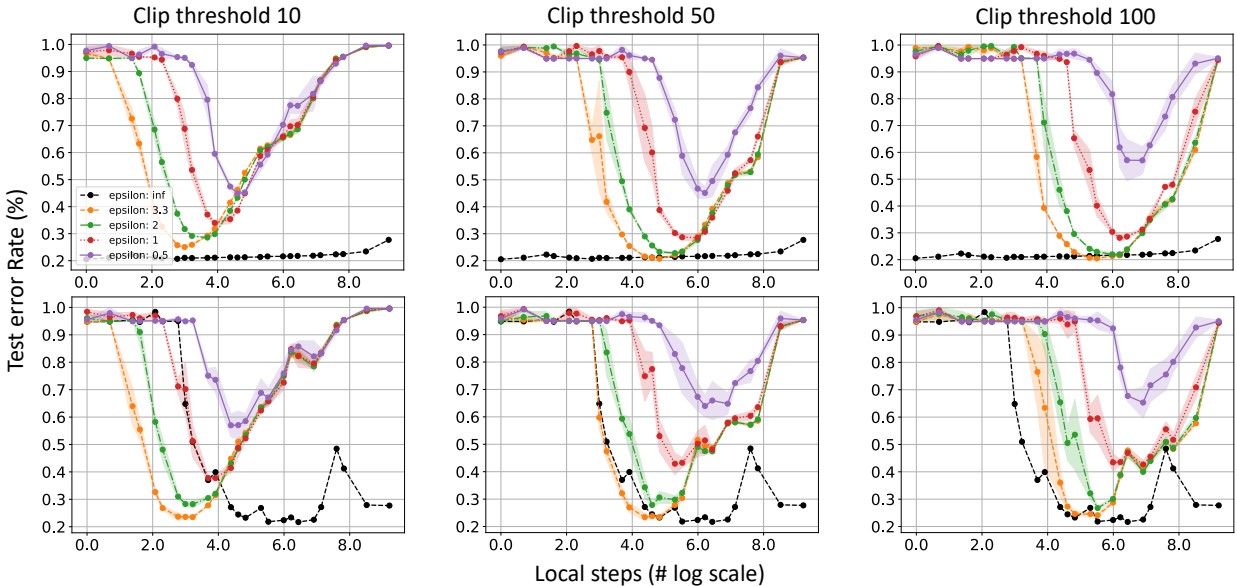

Figure 4: The FEMNIST result. First line: DP-FedAvg, Second line: DP-ScaffNew.

**Observation 3.** *The optimal number of local steps increases and the optimal total iterations number decreases as the privacy degree increases.*

We observe that as the privacy degree increases ($\epsilon$ becomes small), the optimal number of local steps increases and the optimal total iteration number decreases as shown in Figure 2 and 3, respectively. Regarding the number of iterations $T$, in Corollary 1 we prove this corresponding change with $\epsilon$. However, with respect to the phenomenon of local steps, there exists a discrepancy with theory, as the theory shows that it only relates to the $\mu$ and $L$. This might be attributed to inherent complexities in the model.

**Observation 4.** *The optimal local steps depend on the clipping threshold, but it does not significantly impact performance.*

We evaluate the impact of clipping thresholds (at 10, 50, 100) on the local steps for DP federated algorithms to train over FEMNIST in Figure 4. Additional results over other datasets can be found in the appendix. As the clipping threshold increases, the optimal local step number tends to increase but does not impact the test error rate substantially. This is because the increase in $C$ leads to the utility bound equation 2 which becomes dominated by the second term. To minimize this utility bound, $p$ and $T$ must become smaller. This implies the larger expected number of local steps $1/p$.

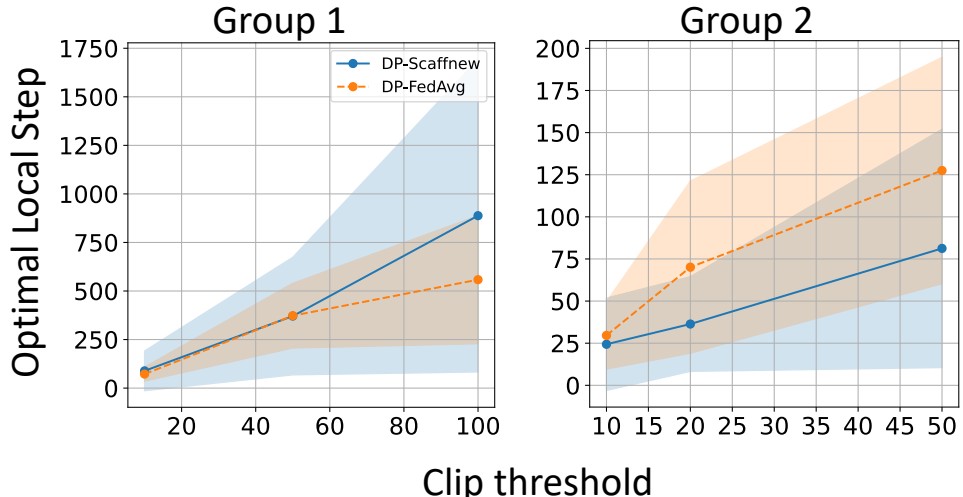

Figure 5: Intra-group optimal local values for two dataset groups: CIFAR10, FEMNIST, and Reddit (Group 1) versus Fed-IXI and Messidor (Group 2).

We further investigate the clipping effect for DP-FedAvg and DP-ScaffNew. We perform this by computing the intra-group mean and variance that are evaluated for four different guarantees for each dataset (where we collect the test error rate against varied local steps), as we show in Figure 5.

After the $R^2$ test, we find that the optimal local steps linearly depend on the clipping threshold for DP-ScaffNew, and on the square root of the clipping threshold for DP-FedAvg. Therefore, the benefit of local steps given the clipping threshold is more significant for DP-ScaffNew than DP-FedAvg. This may be because DP-FedAvg, in contrast to DP-ScaffNew satisfying equation 2, has an additional error term due to data heterogeneity. Also, this observation on the limited benefit of local steps for DP-FedAvg aligns with that for FedAvg by Wang & Joshi (2019).

## 4 Conclusion

This paper shows that DP federated algorithms have the optimal number of local steps and communication rounds to balance privacy and convergence performance. Our theory provides the explicit expression of these hyper-parameters that balance the trade-off between privacy and utility for DP-ScaffNew algorithms. This result holds for strongly convex optimization without requiring data heterogeneity assumptions, unlike existing literature. Extensive experiments on benchmark FL datasets corroborate our findings and demonstrate strong performance for DP-FedAvg and DP-ScaffNew with optimal numbers of local steps and iterations, which are nearly comparable to their non-private counterparts.

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

# A    Benchmark Data Pre-processing

Throughout the experiments, we pre-process FEMNIST, Fed-IXI, Messidor and Reddit before training.

**FEMNIST**    For the FEMNIST dataset, due to its huge size, we randomly sample only 5% out of all the samples before they are split equally among the clients.

**Fed-IXI**    For the Fed-IXI dataset, we follow the same pre-processing steps according to the FLamby software suite Terrail et al. (2022). All scans are geometrically aligned to the MNI template by the NtiftyReg Modat et al. (2014) and re-oriented using ITK to a common space. Finally, based on the whole image histogram, we normalized the intensities and resized them from $83 \times 44 \times 55$ to $48 \times 60 \times 48$.

**Messidor**    For the Messidor dataset, the following pre-processing steps were employed. Initially, black edges were cropped based on a pixel threshold value of 1. Subsequently, every image was resized to a standard dimension of $224 \times 224$. Data augmentation techniques were also integrated, including random horizontal and vertical flips, alongside a restricted random rotation of 10 degrees. We also pre-process its class labels for the binary classification task, according to steps explained by Yan et al. (2023).

**Reddit**    For the Reddit dataset, each text sample is tokenized by using the table provided by LEAF Caldas et al. (2018).

# B    Proofs

## B.1    Proof of Theorem 1

Similarly to (Mishchenko et al., 2022), we first proceed by reformulating equation 1 to

$$\min_{\mathbf{x} \in \mathbb{R}^{nd}} \frac{1}{N} \sum_{i=1}^{N} f_i(x_i) + \phi(\mathbf{x}),$$

where $\mathbf{x} = [x_1, x_2, \ldots, x_N]^\top$ and

$$\phi(\mathbf{x}) = \begin{cases} 0 & \text{if } x_1 = x_2 = \ldots = x_N, \\ \infty & \text{otherwise.} \end{cases}$$

Note that this formulation is equivalent to our original problem equation 1 as it has to be the case $\phi(\mathbf{x}) = 0$ for the optimal solution. Furthermore, note that the gradient step with respect to the first term corresponds to the local step across all the clients, and the proximal step with respect to the second term is equivalent to the averaging of local variables $x_i$'s that corresponds to the communication step. Therefore, this reformulation helps us to better facilitate the theoretical analysis in the federated setup. Furthermore, we would like to highlight that we never use $\phi(\mathbf{x})$ in the analysis. Therefore, our analysis is not affected by this reformulation and the possible functional value of infinity.

In order to solve this equivalent reformulation, we first define useful lifts to $\mathbb{R}^{nd}$ that help us to simplify the analysis. To get better intuition why these lifts might be useful, note that if $x^\star$ is the optimal solution of equation 1 then $\mathbf{x}^\star := \left[ (x^\star)^\top, \ldots, (x^\star)^\top \right]^\top$ is the optimal solution of our reformulation.

First, we define

$$\mathbf{x}_t := \left[ (x_t^{(1)})^\top, \ldots, (x_t^{(N)})^\top \right]^\top,$$

$$\hat{\mathbf{x}}_t := \left[ (\hat{x}_t^{(1)})^\top, \ldots, (\hat{x}_t^{(N)})^\top \right]^\top,$$

$$\tilde{\mathbf{x}}_t := \left[ (x_t)^\top, \ldots, (x_t)^\top \right]^\top,$$

$$\mathbf{x}^\star := \left[ (x^\star)^\top, \dots, (x^\star)^\top \right]^\top,$$

$$\mathbf{g}_t := \left[ (\nabla f_1(x_t^{(1)}))^\top, \dots, (\nabla f(x_t^{(N)}))^\top \right]^\top,$$

$$\mathbf{\Delta}_t := \left[ (\Delta_t^{(1)})^\top, \dots, (\Delta_t^{(N)})^\top \right]^\top,$$

$$\mathbf{h}_t := \left[ (h_t^{(1)})^\top, \dots, (h_t^{(N)})^\top \right]^\top,$$

$$\mathbf{h}^\star := \left[ (\nabla f_1(x^\star))^\top, \dots, (\nabla f_N(x^\star))^\top \right]^\top.$$

Also, let us define the linear operator $\mathbf{A}(\mathbf{y}_t) := \bar{\mathbf{y}}_t$, where $\bar{\mathbf{y}}_t := \left[ (\bar{y}_t)^\top, \dots, (\bar{y}_t)^\top \right]^\top$ and $\bar{y}_t := \frac{1}{N} \sum_{i=1}^N y_t^{(i)}$. By setting $g_i(x) = \nabla f_i(x)$, and $C > \|\hat{x}_{t+1}^{(i)} - x_t\|$ for $i \in \{1, \dots, N\}$ and $t \geq 0$, and by following similar proof arguments as Mishchenko et al. (2022), Algorithm 2 can be expressed equivalently as:

$$\mathbf{x}_{t+1} := \begin{cases} \mathbf{A} \left( \hat{\mathbf{x}}_{t+1} + \mathbf{e}_t \right) & \text{with probability } p \\ \hat{\mathbf{x}}_{t+1} & \text{otherwise} \end{cases}$$

where $\hat{\mathbf{x}}_{t+1} := \mathbf{x}_t - \eta(\mathbf{g}_t - \mathbf{h}_t)$, $\mathbf{h}_{t+1} := \mathbf{h}_t + \frac{p}{\eta}(\mathbf{A}(\mathbf{\Delta}_t) - \mathbf{\Delta}_t)$ and $\mathbf{e}_t \sim \mathcal{N}(0, \sigma^2 I_{N \cdot d})$ since by definition of Algorithm 2, we have with probability $p$

$$\mathbf{x}_{t+1} = \tilde{\mathbf{x}}_t + \mathbf{A}(\mathbf{\Delta}_t) = \tilde{\mathbf{x}}_t + \mathbf{A}(\hat{\mathbf{x}}_{t+1} - \tilde{\mathbf{x}}_t + \mathbf{e}_t) = \mathbf{A}(\hat{\mathbf{x}}_{t+1} + \mathbf{e}_t)$$

as $\mathbf{A}(\tilde{\mathbf{x}}_t) = \tilde{\mathbf{x}}_t$.

Furthermore, note that this can be also written as

$$\mathbf{x}_{t+1} := \begin{cases} \mathbf{A} \left( \hat{\mathbf{x}}_{t+1} - \frac{\eta}{p} \mathbf{h}_t + \mathbf{e}_t \right) & \text{with probability } p \\ \hat{\mathbf{x}}_{t+1} & \text{otherwise} \end{cases}$$

since

$$\mathbf{A}(\mathbf{h}_{t+1}) = \mathbf{A} \left( \mathbf{h}_t + \frac{p}{\eta}(\mathbf{A}(\mathbf{\Delta}_t) - \mathbf{\Delta}_t) \right) = \mathbf{A}(\mathbf{h}_t) + \frac{p}{\eta}(\mathbf{A}(\mathbf{\Delta}_t) - \mathbf{A}(\mathbf{\Delta}_t)) = \mathbf{A}(\mathbf{h}_t)$$

and $\mathbf{A}(\mathbf{h}_0) = 0$. Thus,

$$\mathbf{A}(\mathbf{h}_t) = 0 \text{ for all } t \geq 0. \tag{3}$$

Define $\psi_t := \|\mathbf{x}_t - \mathbf{x}^\star\|^2 + (\eta^2/p^2)\|\mathbf{h}_t - \mathbf{h}^\star\|^2$. Then,

$$\mathbf{E}[\psi_{t+1}] = p\mathbf{E}[T_1 + T_2] + (1-p)\mathbf{E}\left[ \|\hat{\mathbf{x}}_{t+1} - \mathbf{x}^\star\|^2 + \frac{\eta^2}{p^2} \|\mathbf{h}_t - \mathbf{h}^\star\|^2 \right], \tag{4}$$

where

$$T_1 = \|\mathbf{A}(\hat{\mathbf{x}}_{t+1} - \frac{\eta}{p}\mathbf{h}_t + \mathbf{e}_t) - \mathbf{x}^\star\|^2$$

$$= \|\mathbf{A}(\hat{\mathbf{x}}_{t+1} - \frac{\eta}{p}\mathbf{h}_t) + \mathbf{A}(\mathbf{e}_t) - \mathbf{A}(\mathbf{x}^\star - \frac{\eta}{p}\mathbf{h}^\star)\|^2$$

$$= \|\mathbf{A}(\hat{\mathbf{x}}_{t+1} - \frac{\eta}{p}\mathbf{h}_t) - \mathbf{A}(\mathbf{x}^\star - \frac{\eta}{p}\mathbf{h}^\star)\|^2 + \|\mathbf{A}(\mathbf{e}_t)\|^2$$

since $\mathbf{x}^\star = \mathbf{A}(\mathbf{x}^\star) = \mathbf{A}(\mathbf{x}^\star - \frac{\eta}{p}\mathbf{h}^\star)$ and $\mathbf{e}_t$ is independent noise, and

$$
\begin{aligned}
T_2 &= \left\| \frac{\eta}{p}(\mathbf{h}_t - \mathbf{h}^\star) + (\mathbf{A}\left(\boldsymbol{\Delta}_t\right) - \boldsymbol{\Delta}_t) \right\|^2 \\
&= \left\| \frac{\eta}{p}(\mathbf{h}_t - \mathbf{h}^\star) + (\mathbf{A}(\hat{\mathbf{x}}_{t+1} - \tilde{\mathbf{x}}_t + \mathbf{e}_t) - (\hat{\mathbf{x}}_{t+1} - \tilde{\mathbf{x}}_t + \mathbf{e}_t)) \right\|^2 \\
&= \left\| \frac{\eta}{p}(\mathbf{h}_t - \mathbf{h}^\star) + (\mathbf{A}(\hat{\mathbf{x}}_{t+1} + \mathbf{e}_t) - (\hat{\mathbf{x}}_{t+1} + \mathbf{e}_t)) \right\|^2 \\
&= \| \mathbf{A}(\hat{\mathbf{x}}_{t+1} - \tfrac{\eta}{p}\mathbf{h}_t) - (\hat{\mathbf{x}}_{t+1} - \tfrac{\eta}{p}\mathbf{h}_t) - \mathbf{A}(\mathbf{x}^\star - \tfrac{\eta}{p}\mathbf{h}^\star) + (\mathbf{x}^\star - \tfrac{\eta}{p}\mathbf{h}^\star) \|^2 + \| \mathbf{A}(\mathbf{e}_t) - \mathbf{e}_t \|^2
\end{aligned}
$$

since $\mathbf{x}^\star = \mathbf{A}(\mathbf{x}^\star) = \mathbf{A}(\mathbf{x}^\star - \frac{\eta}{p}\mathbf{h}^\star)$, $\mathbf{A}\left(\mathbf{h}_t\right) = 0$, and $\mathbf{e}_t$ is independent noise.

Let us define $\mathbf{x} := \hat{\mathbf{x}}_{t+1} - \frac{\eta}{p}\mathbf{h}_t$ and $\mathbf{y} := \mathbf{x}^\star - \frac{\eta}{p}\mathbf{h}^\star$. Since $\langle \mathbf{A}(\mathbf{z}), \mathbf{z} \rangle = \|\mathbf{A}(\mathbf{z})\|^2$ for all $\mathbf{z} \in \mathbb{R}^{Nd}$, we have

$$
\begin{aligned}
\mathbf{E}[T_1 + T_2] &= \mathbf{E}\left\| \mathbf{A}(\mathbf{x}) - \mathbf{A}(\mathbf{y}) \right\|^2 + \mathbf{E}\left\| [\mathbf{A}(\mathbf{x}) - \mathbf{x}] - [\mathbf{A}(\mathbf{y}) - \mathbf{y}] \right\|^2 + \mathbf{E}\| \mathbf{A}(\mathbf{e}_t) - \mathbf{e}_t \|^2 + \mathbf{E}\| \mathbf{A}(\mathbf{e}_t) \|^2 \\
&= \mathbf{E}\left\| \mathbf{A}(\mathbf{x} - \mathbf{y}) \right\|^2 + \mathbf{E}\left\| \mathbf{A}(\mathbf{x} - \mathbf{y}) - (\mathbf{x} - \mathbf{y}) \right\|^2 + \mathbf{E}\| \mathbf{A}(\mathbf{e}_t) - \mathbf{e}_t \|^2 + \mathbf{E}\| \mathbf{A}(\mathbf{e}_t) \|^2 \\
&= \mathbf{E}\| \mathbf{x} - \mathbf{y} \|^2 + \mathbf{E}\| \mathbf{e}_t \|^2.
\end{aligned}
$$

Let us define $\mathbf{y}_t := [\hat{\mathbf{x}}_{t+1} - \mathbf{x}^\star] - \frac{\eta}{p}[\mathbf{h}_t - \mathbf{h}^\star]$. Therefore,

$$
\mathbf{E}[T_1 + T_2] \le \mathbf{E}\| \mathbf{y}_t \|^2 + Nd\sigma^2.
$$

Plugging the upper-bound for $\mathbf{E}[T_1 + T_2]$ into equation 4 thus yields

$$
\mathbf{E}[\psi_{t+1}] \le p\mathbf{E}\| \mathbf{y}_t \|^2 + (1-p)\mathbf{E}\left[ \| \hat{\mathbf{x}}_{t+1} - \mathbf{x}^\star \|^2 + \frac{\eta^2}{p^2} \| \mathbf{h}_t - \mathbf{h}^\star \|^2 \right] + pNd\sigma^2.
$$

Next, by the fact that $\|x - y\|^2 = \|x\|^2 - 2\langle x, y \rangle + \|y\|^2$ with $x := \hat{\mathbf{x}}_{t+1} - \mathbf{x}^\star$ and $y := \frac{\eta}{p}[\mathbf{h}_t - \mathbf{h}^\star]$, and that $\hat{\mathbf{x}}_{t+1} := \mathbf{x}_{t+1} - \eta(\mathbf{g}_t - \mathbf{h}_t)$,

$$
\begin{aligned}
\mathbf{E}[\psi_{t+1}] &\le \mathbf{E}\left[ \| \hat{\mathbf{x}}_{t+1} - \mathbf{x}^\star \|^2 + \frac{\eta^2}{p^2} \| \mathbf{h}_t - \mathbf{h}^\star \|^2 \right] - 2\eta\mathbf{E}\langle \hat{\mathbf{x}}_{t+1} - \mathbf{x}^\star, \mathbf{h}_t - \mathbf{h}^\star \rangle + pNd\sigma^2 \\
&= \mathbf{E}\| \hat{\mathbf{x}}_{t+1} - \mathbf{x}^\star - \eta[\mathbf{h}_t - \mathbf{h}^\star] \|^2 + (1 - p^2)\frac{\eta^2}{p^2}\mathbf{E}\| \mathbf{h}_t - \mathbf{h}^\star \|^2 + pNd\sigma^2 \\
&= \mathbf{E}\| \mathbf{x}_t - \mathbf{x}^\star - \eta[\mathbf{g}_t - \mathbf{h}^\star] \|^2 + (1 - p^2)\frac{\eta^2}{p^2}\mathbf{E}\| \mathbf{h}_t - \mathbf{h}^\star \|^2 + pNd\sigma^2.
\end{aligned}
$$

If each $f_i(x)$ is $\mu$-strongly convex and $L$-smooth, then for $\eta \le 1/L$

$$
\begin{aligned}
\mathbf{E}[\psi_{t+1}] &\le (1 - \mu\eta)\mathbf{E}\| \mathbf{x}_t - \mathbf{x}^\star \|^2 + (1 - p^2)\frac{\eta^2}{p^2}\mathbf{E}\| \mathbf{h}_t - \mathbf{h}^\star \|^2 + 2p\sigma^2 \\
&\le \rho\mathbf{E}[\psi_t] + pNd\sigma^2.
\end{aligned}
$$

where $\rho := \max(1 - \mu\eta, 1 - p^2)$.

If $p \in (0, 1]$, then applying this inequality recursively over $t = 0, 1, \ldots, T-1$ yields

$$
\mathbf{E}[\psi_T] \le \rho^T \psi_0 + \frac{pNd}{1-\rho}\sigma^2.
$$

Finally, by letting the privacy variance $\sigma^2 = pv\frac{C^2 T \ln(1/\delta)}{\epsilon^2}$ for $\epsilon \le uT$ and $\delta, B > 0$ according to Lemma 1, we complete the proof.

## B.2 Proof of Corollary 1

If $\eta = 1/L$, then $\max(1 - \mu\eta, 1 - p^2) = 1 - \min(\mu/L, p^2) := 1 - \theta$. From Theorem 1 we have

$$\mathbf{E}[\psi_T] \le (1 - \theta)^T \psi_0 + \frac{p^2 N d}{\theta} \cdot \frac{v C^2 T \ln(1/\delta)}{\epsilon^2}.$$

Note that $(1 - \theta)^T \le (1 - \mu/L)^T$, while $p^2/\theta = \max\left(p^2 L/\mu, 1\right) \ge 1$. We hence minimize the convergence bound by letting $p^\star = \operatorname{argmin}_p \frac{p^2}{\theta} = \sqrt{\frac{\mu}{L}}$, which yields

$$\mathbf{E}[\psi_T] \le \left(1 - \frac{\mu}{L}\right)^T \psi_0 + \frac{v C^2 N d T \ln(1/\delta)}{\epsilon^2}.$$

Next, we find the optimal number of iterations $T^\star$ such that

$$T^\star = \operatorname{argmin}_T \left(1 - \frac{\mu}{L}\right)^T \psi_0 + \frac{v C^2 N d T \ln(1/\delta)}{\epsilon^2}.$$

Since $x = \exp\left(\ln(x)\right)$,

$$T^\star = \operatorname{argmin}_T \ B(T)$$
$$:= \exp\left(T \ln\left(1 - \frac{\mu}{L}\right)\right) \psi_0 + \frac{v C^2 N d T \ln(1/\delta)}{\epsilon^2}.$$

Therefore,

$$\frac{d}{dT} B(T) = \left(1 - \frac{\mu}{L}\right)^T \psi_0 \cdot \ln\left(1 - \frac{\mu}{L}\right) + \frac{v C^2 N d \ln(1/\delta)}{\epsilon^2}, \qquad \text{and}$$
$$\frac{d^2}{dT^2} B(T) = \left(1 - \frac{\mu}{L}\right)^T \psi_0 \cdot \ln\left(1 - \frac{\mu}{L}\right)^2.$$

Since $\frac{d^2}{dT^2} B(T) > 0$ for all $T \ge 1$, $T^\star = \operatorname{argmin}_T B(T)$ can be found by setting $\frac{d}{dT} B(T) = 0$ which yields

$$\left(1 - \frac{\mu}{L}\right)^{T^\star} \psi_0 \cdot \ln\left(1 - \frac{\mu}{L}\right) + \frac{v C^2 N d \ln(1/\delta)}{\epsilon^2} = 0.$$

Finally, by using the fact that $\ln(1 - \mu/L) = -\ln([1 - \mu/L]^{-1})$ and by re-arranging the terms, we complete the proof.

# C Additional Results

We use the same problem and hyper-parameter settings as Table 1. We present different clip threshold results and more dataset results in this section.

## C.1 Local Client Update Norm

We find that the correct shifts ensure that not only does the global update converge to zero, but also the local updates diminish. Consequently, the clipping becomes active only in the initial rounds. For overparametrized models, where the optimal shifts are zeros, even FedAvg exhibits this property.

Specifically, we observed that for individual clients—not on average—it typically takes less than 5 communication rounds for the norm of the updates to fall below the clipping threshold. This phenomenon is concretely illustrated in Figure 11 for the first client in the Fed-IXI dataset, where we track the gradient norms across individual clients.

Additionally, we compare loss curves with and without gradient clipping. These results suggest that particularly in the minimal clipping threshold settings (i.e., the most challenging) of our study, the impact of clipping on the updates of individual clients is minimal.

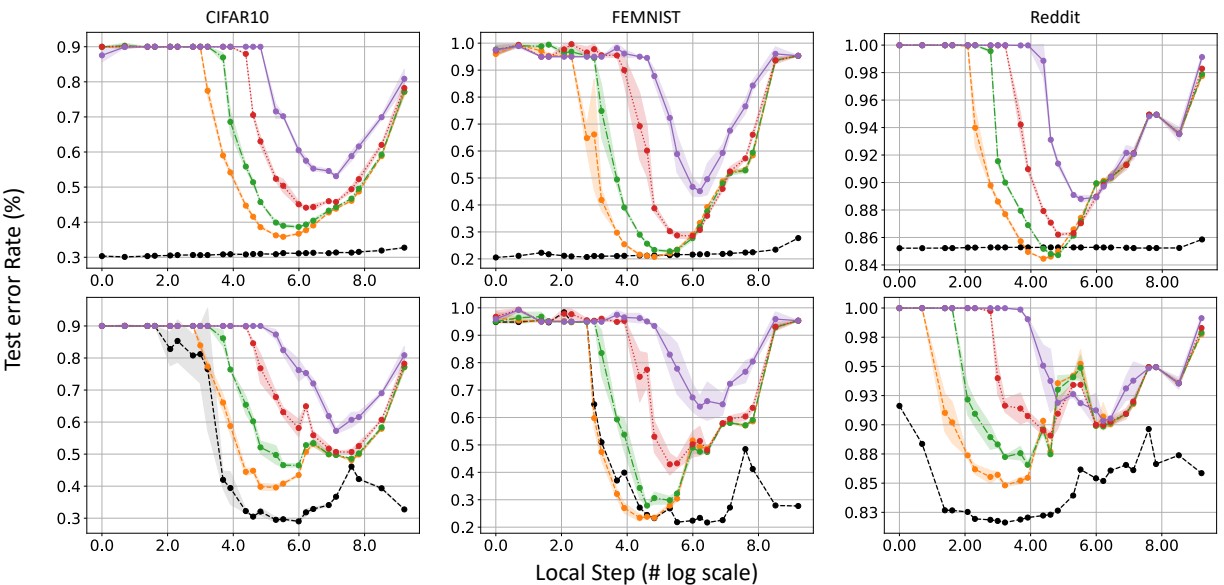

Figure 6: First line: DP-FedAvg, Second line: DP-ScaffNew. Clip threshold: 50.

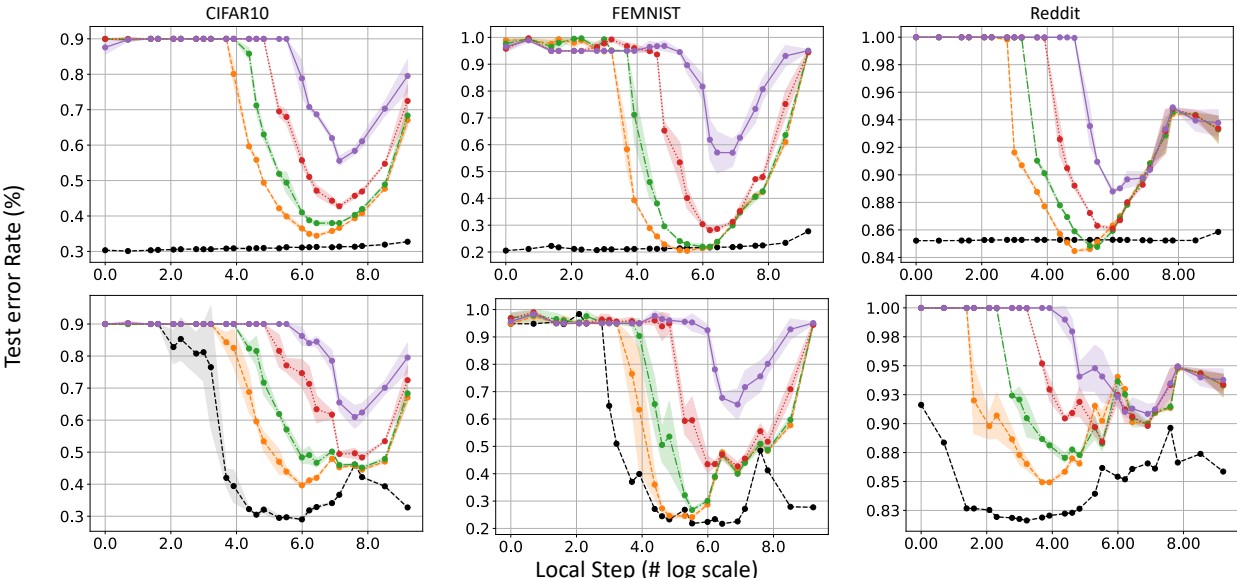

Figure 7: First line: DP-FedAvg, Second line: DP-ScaffNew. Clip threshold: 100.

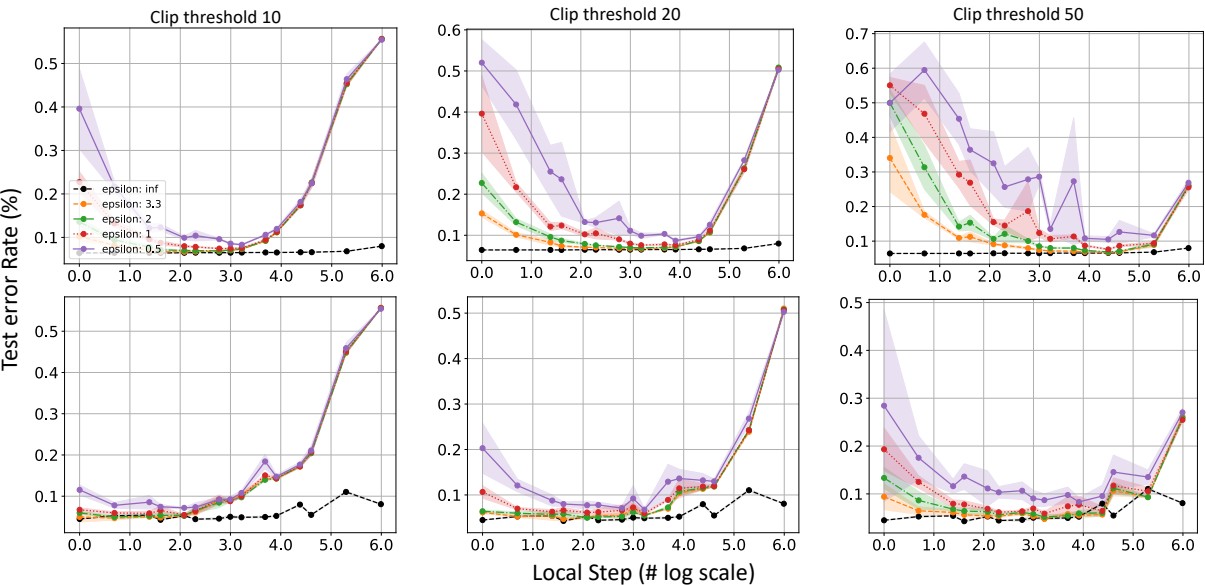

Figure 8: First line: DP-FedAvg, Second line: DP-ScaffNew. Fed-IXI result with fixed iteration times.

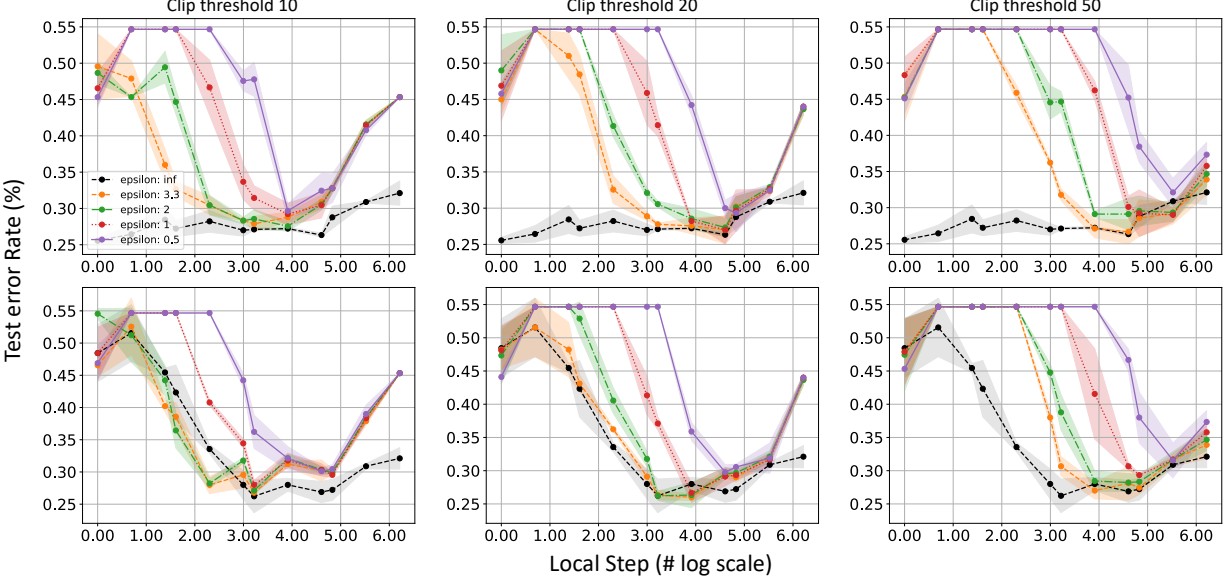

Figure 9: First line: DP-FedAvg, Second line: DP-ScaffNew. Messidor result with fixed iteration times.

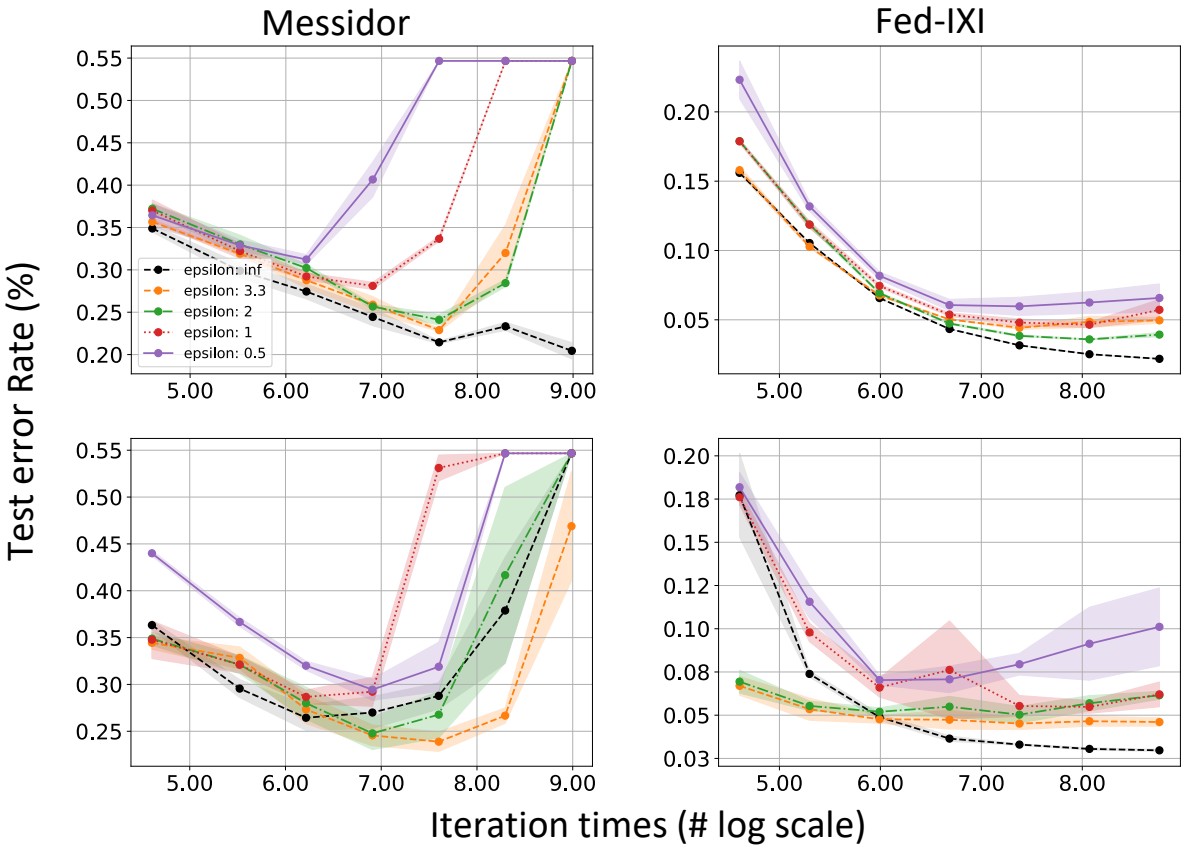

Figure 10: First line: DP-FedAvg, Second line: DP-ScaffNew. Clip threshold: 10. All local step fix to the optimal of its privacy budget.

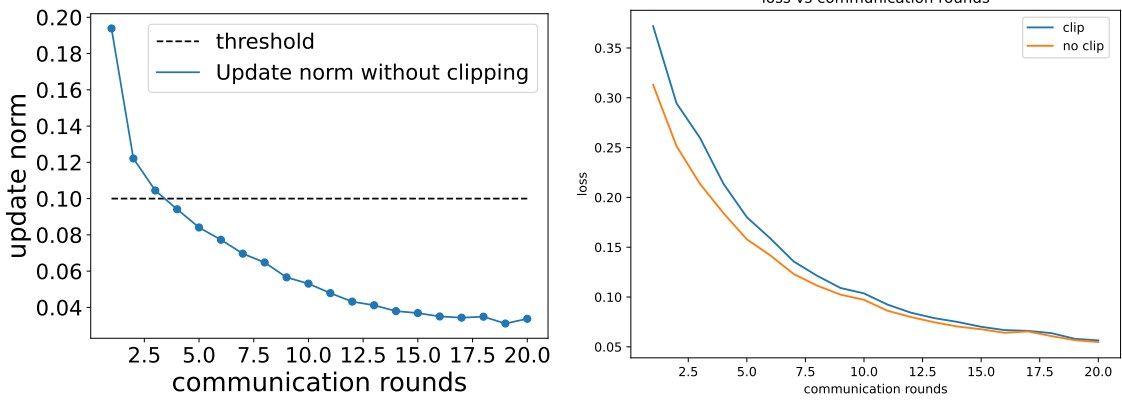

Figure 11: Left. Client 1's update tensor norm. Right. Loss curve for Client 1 with and without clip

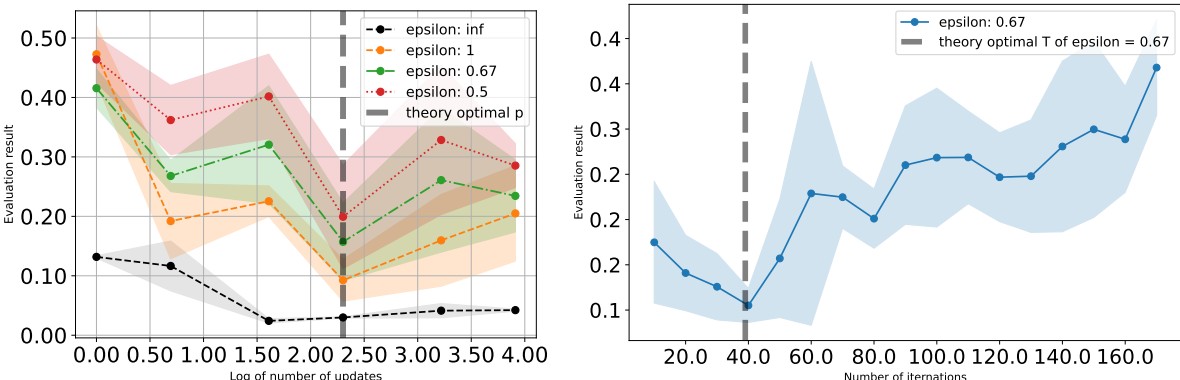

Figure 12: Optimal number of local steps and iteration times of Mushroom dataset

# D  Logistic regression

We use strongly convex logistic regression with closed-formed formulas for strong convexity and smoothness parameters to mimic our theoretical setup. We experiment with the mushroom dataset (Chang & Lin, 2011).

Firstly, we showcase that our theoretical predictions for the optimal number of local steps align with what the theory suggests; see Figure 12. Furthermore, we can see that the optimal value of the local step is independent of the $\epsilon$ as predicted by our theory. Finally, we compare the prediction of the optimal number of steps $T^{\star}$ with the experiments. As we display in Figure 12, our theory is able to predict this value with high precision.

