# OpenReview forum: "Balancing Privacy and Performance for Private Federated Learning Algorithms"
_TMLR — Rejected by TMLR_

### Review · Reviewer_24ts · 2023-11-07

**Summary Of Contributions:**

This paper studies what is the optimal values of the parameter T (number of iterations) for a differentially private optimization algorithm.

**Audience:**

Yes

**Broader Impact Concerns:**

No ethical concerns.

**Claims And Evidence:**

No

**Requested Changes:**

As pointed out, at several places the text is not correct in its current formulation.  Each of these points need to be corrected or explained, and the text improved accordingly.

The only real contribution I see in this paper is the formula for the optimal parameter value $T^*$.  Therefore, it seems this (whether the formula of $T^*$ privides a good estimate of the optimum in practice) is the point experiments should study.  Possibly, improving Eq (2) may lead to better estimates of $T^*$.  The paper is unsuitable for publication (does not meet the first criterion "claims and evidence") without such experiments.

**Strengths And Weaknesses:**

The question of what are the best parameter values for optimization algorithms is interesting.  Here we have the quite common situation where a parameter has an improving and deteriorating impact, and somewhere in the middle is the best spot.  The paper therefore optimizes Eq (2) for T, which is a normal strategy.  However, I'm not convinced that Eq (2) is an optimal equation, i.e., it looks like a rather loose bound.  While it is possible to optimize a loose bound, the problem is that the optimal point in reality may be quite different from the optimal point on the loosely bounding function.
The experiments show that an optimal point exists, which is not very surprising.  It would be much more interesting if the experiments would carefully evaluate how accurate the obtained formula for the optimal $T^*$ is.

The text is well written, well organized and understandable.
At several places, the arguments made are not fully sound.

A good number of references to related work is provided, even though for some results alternative articles exist which provide similar conclusions (e.g., the presented algorithm is very similar to DP-SGD and its variants, which isn't mentioned here).



Some details:

The choice of ScaffNew is motivated as "since FedAvg reaches incorrect stationary points ... ScaffNew Mishchenko et al. (2022) that eliminates this issue", but I doubt ScaffNew really addresses that issue.  No evidence is provided.

As Algorithm 2 is written, it seems each client independently tosses a coin and decides whether to send its model update to the central server.  However, the THEN branch also computes $x_{t+1}$ from $x_t$ and the local model updates.  I guess that should only happen once, not for every client (i.e., it should be outside the loop).  Else, some clients may decide (based on their coin toss) that $x_{t+1}=x_t$ and others may decide that $x_{t+1} = x_t + (1/N) \sum \Delta_t^{(i)}$.
The text refers to lines in Algorithm 2: "DP-ScaffNew adds the extra correction term $h_t^{(i)}$ (line 5, 8 and 13, Alg. 2)", while Algorithm 2 has no line numbering.

It is unclear why assumption (A) is not relevant if the model converges to a zero gradient.  Isn't the goal of optimization always to converge to an optimum with zero gradient (unless there are constraints) ?

The following statement is false: "However, we show that as the algorithm converges, the norms of the updates decrease, and clipping is only active for the first few rounds.".  In particular, not the average gradient is clipped but the individual gradients of individual samples.  Hence, it is well possible that the average gradient is zero (and an optimum is reached) while multiple individual gradients on instances are not zero, to the extent clipping is still active.

In Eq (2), superscript T seems to be exponentiation in $\theta^T$ (as $\theta$ is a real number), while in the line following Eq (2), superscript $T$ denotes a vector/matrix transpose.  Please disambiguate notations, e.g., by using $v^\top$ for transpose and $\theta^T$ for exponentiation.

It feels somewhat unsatisfactory that Eq (2) depends linearly on T.  In fact, without any noise, the distance of x_T to x^* decreases exponentially in T.  I would hope that if in the first iteration one adds noise, the impact of that noise (added distance to x^*) also decreases exponentially (with the same rate) as more iterations happen (T increases).

The proofs in appendix can be made more rigorous, and more explanatory text would benefit understandability.

---

> ### Author Response · Authors · 2023-12-15
> **Author's Response**
>
> We would like to thank the reviewer for their valuable feedback and time spent reviewing our manuscript. We appreciate that the reviewers found many strong points in our manuscript, including:
> - The question of what the best parameter values for optimization algorithms are is interesting.
> - The manuscript is well-written, well-organized, and understandable.
>
> We address all the reviewer's concerns below:
>
> > A good number of references to related work is provided, even though for some results, alternative articles exist which provide similar conclusions (e.g., the presented algorithm is very similar to DP-SGD and its variants, which isn't mentioned here).
>
> We appreciate your comment regarding the similarities to DP-SGD and its variants. We agree that this is the case and our construction basically consists of combining ScaffNew with the Gaussian mechanism, as is the typical way to extend federated learning algorithms to satisfy DP. We would like to point out that the new algorithm is not the main contribution of our work, but the analysis reveals that there is an optimal number of local steps and communication rounds each client should take for solving strongly convex problems.
>
> >  I doubt ScaffNew really addresses that issue. (FedAvg reaches incorrect stationary points)
>
> Let us clarify below. It is indeed the case that FedAvg reaches incorrect stationary points for any fixed step size. For instance, this is characterized via the fixed point method characterization in Theorem 2.11 of Malinovskiy et al., 2020 (https://proceedings.mlr.press/v119/malinovskiy20a/malinovskiy20a.pdf), where the authors precisely define exact linear convergence of FedAvg to the incorrect solution, i.e., not the optimal solution of the original objective. Intuitively, without control variate adjustment, the fixed point drifts away from the optimal solution. Therefore, ProxSkip fixes this issue by correcting local steps via learnable shift, making the optimal solution also the fixed point solution. This is the reason why we built our method on top of the ScaffNew (i.e., ProxSkip) method. We can add this detailed description to the appendix if the reviewer thinks this would benefit our presentation.
>
> > As Algorithm 2 is written, it seems each client independently tosses a coin and decides whether to send its model update to the central server. However, the THEN branch also computes $x_{t+1}$ from $x_{t}$ and the local model updates. I guess that should only happen once, not for every client (i.e., it should be outside the loop). Else, some clients may decide (based on their coin toss) that $x_{t+1} = x_t$ and others may decide that $x_{t+1} = x_t + (1/N) \sum_{i=1}^N \Delta^{(i)}_t$. The text refers to lines in Algorithm 2: "DP-ScaffNew adds the extra correction term $h^{(i)}_t$ (line 5, 8 and 13, Alg. 2)", while Algorithm 2 has no line numbering.
>
> Thank you for this insightful comment; this is indeed the case. We have fixed this typo and clarified that the coin toss is global.
>
> > It is unclear why assumption (A) is not relevant if the model converges to a zero gradient. Isn't the goal of optimization always to converge to an optimum with zero gradient (unless there are constraints)?
>
> Thank you for your comment, and we apologize for any confusion caused. We understand your concern regarding our approach to achieving a zero full gradient and its implications for reaching a global minimum in a convex function. To clarify, our comment in the manuscript refers to the condition where individual gradients $g_i(x^\star)$ at the optimum are zero, indicating the absence of extra noise due to stochastic gradients. This condition not only preserves the fixed point property of the optimal solution but also allows us to avoid relying on full local gradients theoretically. In this case, our method can be applied to stochastic local gradients while maintaining all other results.
>
> It's important to note that the full gradient averages these individual gradients. There may be scenarios where these individual gradients are non-zero, yet their average—the full gradient—is zero. To address this, we introduced assumption (A) in our manuscript, which helps avoid such scenarios.

---

> > ### Author Response · Authors · 2023-12-15
> > **Author's Response Cont.**
> >
> > >  The following statement is false: "However, we show that as the algorithm converges, the norms of the updates decrease, and clipping is only active for the first few rounds.". In particular, not the average gradient is clipped but the individual gradients of individual samples. Hence, it is well possible that the average gradient is zero (and an optimum is reached) while multiple individual gradients on instances are not zero, to the extent clipping is still active.
> >
> > Thank you for this important comment. Your observation is indeed insightful. This is precisely why we propose adapting the ProxSkip shift framework in combination with differential privacy. The correct shifts ensure that not only does the global update converge to zero, but also the local updates diminish. Consequently, the clipping becomes active only in the initial rounds, as we argue. Furthermore, for overparametrized models, where the optimal shifts are zeros, even FedAvg exhibits this property. We have also verified this empirically. Specifically, we observed that for individual clients—not on average—it typically takes only one or two communication rounds for the norm of the updates to fall below the clipping threshold. This phenomenon is concretely illustrated in the figures below for the first client in the Fed-IXI dataset, where we track the gradient norms across individual clients. Additionally, we compare loss curves with and without gradient clipping. These results suggest that, particularly in the minimal clipping threshold settings of our study, the impact of clipping on the updates of individual clients is minimal. This clarification provides a more accurate context to our initial statement and aligns with the experimental evidence presented in our manuscript. We have added the extra experiment to the appendix.
> >
> > > Notation issues.
> >
> > Thank you for spotting this issue. We have removed this ambiguity and followed the reviewer’s suggestion.
> >
> > > It feels somewhat unsatisfactory that Eq (2) depends linearly on $T$. In fact, without any noise, the distance of $x_T$ to $x^*$ decreases exponentially in $T$. I would hope that if, in the first iteration, one adds noise, the impact of that noise (added distance to $x^*$) also decreases exponentially (with the same rate) as more iterations happen ($T$ increases).
> >
> > Thank you for raising this important point about the linear dependence on $T$ in Equation (2)—this scaling results from incorporating noise at each communication round to ensure differential privacy (DP) guarantees. The $pT$ scaling arises from applying the moments' accountant composition lemma introduced by Abadi et al. (2016), which, based on our current understanding, is the tightest composition lemma available for such analyses.
> >
> > As we increase the number of iterations $T$, more noise is inherently required to maintain DP guarantees. This necessity results in the second term of the equation deteriorating linearly with  $T$. In the context of differential privacy, this trade-off is a natural consequence of balancing privacy with convergence.
> >
> > Our methodology and the resulting equation reflect this intrinsic challenge in DP-ensured machine learning. Further advancements in DP research may provide more efficient ways to reduce the impact of noise over iterations. Still, as of now, the $pT$ scaling remains a fundamental aspect of ensuring robust privacy protections in our framework.
> >
> > > The proofs in the appendix can be made more rigorous, and more explanatory text would benefit understandability.
> >
> > Thank you. We have added more intuition about the proof technique to the appendix and more accompanying text. Furthermore, we have fixed some issues with the notation in the proofs. Please see the updated proof in the appendix.
> >
> > > The only real contribution I see in this paper is the formula for the optimal parameter value $T^*$. Therefore, it seems this (whether the formula of provides a good estimate of the optimum $T^*$ in practice) is the point experiments should study. Possibly, improving Eq (2) may lead to better estimates of $T^*$. The paper is unsuitable for publication (it does not meet the first criterion "claims and evidence") without such experiments.
> >
> > Thank you for this comment; we have added the experiment as required by the reviewer. Concretely, we use strongly convex logistic regression with closed-formed formulas for strong convexity and smoothness parameters to mimic our theoretical setup. We experiment with the mushroom dataset. Firstly, we showcase that our theoretical predictions for the optimal number of local steps align with what the theory suggests. Furthermore, we can see that the optimal value of the local step is independent of the $\epsilon$ as predicted by our theory. Finally, we compare the prediction of the optimal number of steps $T^\star$ with the experiments. As we display in Figure 12 (in the appendix), our theory is able to predict this value with high precision.

---

> > > ### Comment · Reviewer_24ts · 2023-12-16
> > >
> > > >>    The following statement is false: "However, we show that as the algorithm converges, the norms of the updates decrease, and clipping is only active for the first few rounds.". In particular, not the average gradient is clipped but the individual gradients of individual samples. Hence, it is well possible that the average gradient is zero (and an optimum is reached) while multiple individual gradients on instances are not zero, to the extent clipping is still active.
> > >
> > > > Thank you for this important comment. Your observation is indeed insightful. This is precisely why we propose adapting the ProxSkip shift framework in combination with differential privacy. The correct shifts ensure that not only does the global update converge to zero, but also the local updates diminish. Consequently, the clipping becomes active only in the initial rounds, as we argue. Furthermore, for overparametrized models, where the optimal shifts are zeros, even FedAvg exhibits this property. We have also verified this empirically. Specifically, we observed that for individual clients—not on average—it typically takes only one or two communication rounds for the norm of the updates to fall below the clipping threshold. This phenomenon is concretely illustrated in the figures below for the first client in the Fed-IXI dataset, where we track the gradient norms across individual clients. Additionally, we compare loss curves with and without gradient clipping. These results suggest that, particularly in the minimal clipping threshold settings of our study, the impact of clipping on the updates of individual clients is minimal. This clarification provides a more accurate context to our initial statement and aligns with the experimental evidence presented in our manuscript. We have added the extra experiment to the appendix.
> > >
> > > I'm afraid I don't follow this answer.
> > > You refer in your answer to ProxSkip and "shifts".  The paper only contains once the word "shift" at page 4 and twice on page 16.  At page 4 the paper says:
> > >
> > > > However, since FedAvg reaches incorrect stationary points Pathak & Wainwright (2020), we rather consider  the DP version of ScaffNew Mishchenko et al. (2022) that eliminates this issue by adding an extra drift/shift to the local gradient.
> > >
> > > However, [Mishchenko et al. (2022)] does not say much about privacy.  Algorithm 2 is then the DP version, but it is unclear why Algorithm 2 wouldn't suffer from the same problem as I outlined above: even if the global model has converged, the local gradients g_i won't be zero, may even be large.  Then, skipping communication until \theta_t = 1 cumulates these non-zero gradients locally, before clipping.  However, there doesn't seem to be a strong argument why these cumulated gradients won't reach the clipping threshold.  So what we would need to establish is that IF we are in the situation where the global model has converged, i.e., does not change anymore except small variations due to the randomness of the algorithm, THEN no clipping should occur anymore.  Can you point me to where you prove this?

---

> ### Author Response · Authors · 2023-12-16
> **Shift for client $i$ = $h_t^{(i)}$**
>
> Thank you for this comment!
>
> Please let us clarify what we mean by shifts, and we apologize for the confusion.
>
> In our response, we refer to $h_t^{(i)}$ as shifts. Please note that if we initialize $x^0 = x^*$ and $h_0^{(i)} = \nabla f_i(x^*)$, then in each local step of DP-ScaffNew, we will not move as $\nabla f_i(x^{(i)}_t) - h_t^{(i)} = 0$ even if we skip communication. Furthermore, note that this is the setup to which our method converges, as we show in Section B.1, due to our definition of $\psi_t$ that converges to zero up to neighborhood due to noise. Therefore, what the reviewer describes will not happen in the setup that we analyze since it might be the case $\nabla f_i(x^{(i)}_t)$ is non-zero, but it will be eventually canceled by $h_t^{(i)}$ as the model converges.
>
> Please let us know whether this addresses your concern.

---

> > ### Comment · Reviewer_24ts · 2023-12-17
> >
> > ok, so let's assume with $x^0$ you mean $x_0$.
> > Initializing $x_0=x^*$ is exactly the point I want to understand.  Let's assume for now that also $h_0=\nabla f_i(x^*)$ even though this is not what line 1 of Algorithm 2 says.
> >
> > Indeed, then $g_i(x_0^{(i)})-h_i^{(i)} = 0$.  If no communication happens, then $x_{t+1}=x_t$.  However, if communication happens, we get to line 8.  We know that $p$ is small, so $\eta/p$ may be somewhat large.  ${\hat{x}}_{t+1}^{(i)}-\frac{\eta}{p}h_t^{(i)}-x_t$ will then be equal to $-\frac{\eta}{p}h_t^{(i)}$  and may get clipped if the gradients on the points in the dataset are sufficiently different, which is possible.  As a result, even if we initialize $x_0=x^*$ and ignore noise, the algorithm may drift away from the optimal solution $x^*$ as soon as communication happens (if the average of the clipped gradients does not equal to average of the real gradients which is 0).  Even without noise, $x^*$ (with the $h_i$ you propose) does not seem a stationary point.

---

> > > ### Author Response · Authors · 2023-12-18
> > > **Thank you for the detailed comment.**
> > >
> > > Thank you for the detailed comment. It is indeed the case that what the reviewer mentions might happen if gradients at optimum are sufficiently different. However, this is easy to address and consistent with our implementation, as discussed below.
> > >
> > > Firstly, note that, we do not require $h_0^{(i)} = \nabla f_i(x^\star)$ but our analysis guarantees $h_t^{(i)} \to \nabla f_i(x^\star)$ as $t \to \infty$, i.e., as $x_t \to x^\star$.
> > >
> > > Regarding clipping a large value due to sufficient difference in local gradients in optimum, the reviewer is correct in their reasoning. To fix this issue, we have updated line 8 of Algorithm 2 by removing the term $\frac{\eta}{p} h_t^{(i)}$, which is consistent with how we run our experiments. Please note that this is also consistent with our theory since, in theory, we do not consider clipping, and by construction, it is always the case that $\sum_{i=1}^N h_t^{(i)} = 0$. Thus, line 9 (Global averaging) remains the same regardless of whether we include the term $\frac{\eta}{p} h_t^{(i)}$ in line 8 or not.
> > >
> > > With this change, DP-ScaffNew with clipping is guaranteed not to escape the optimal solution if we initialize all parameters as discussed in our prior response and do not consider noise since we always clip zero value.
> > >
> > > Please let us know if this answers your concern.

---

> > > > ### Comment · Reviewer_24ts · 2023-12-18
> > > >
> > > > Your change in line 8 introduces a new problem:  Consider any $x_t$ and $h_t^{(i)} = \nabla f_i(x_t)$ (similarly as above but now not for $x^*$ but for a different point $x^C$.  Then, $x^C$ becomes a stationary point too.  In particular, with $x_t = x^C$ and $h_t^{(i)}(x_t)=\nabla f_i(x_t)$ we have again $g_i(x_t^{(i)})-h_i = 0$ and in line 6 we get ${\hat{x}}_{t+1} = x_t^{(i)}$.  Then, Line 7 gives $\Delta_t^{(i)}=0$ and Lines 9 and 13 seem to keep $h_{t+1}^{(i)}=h_t^{(i)}$ unchanged.
> > > >
> > > > The reviewer (nor the reviewer) starts reading the appendix without incentive.  Please write clearly in the text next to Theorem 1 where the proof can be found.
> > > >
> > > > There are several concerns with this proof of Theorem 1:
> > > >
> > > > * It is a sequence of formal computations without many sentences providing the reader with an intuition, so it is hard to verify the individual steps are correct.
> > > > * The proof of Theorem 1 still says that in case of communication, $x_{t+1}= A(x_t -\frac{\eta}{p} h_t + e_t)$, i.e., without the change in line 8.
> > > > * The definition of $\phi(x)$ introduces a possibility of getting $\infty$.  When using $\infty$ a lot of the computation may get a different meaning, and hence everywhere in the proof we must be very careful to also check what happens in that case
> > > > * It is strange that you first prove $\mathbb{E}[T_1+T_2] = \mathbb{E}[ \|y_t\|^2 + \sigma^2$ and then]need a complex argument to show the trivial consequence that $\mathbb{E}[T_1+T_2] \le \mathbb{E}[ \|y_t\|^2 + 2\sigma^2$.  Maybe I miss something or there is some typing mistake.
> > > >
> > > > The theorem seems to try to show that $\|x^*- x_t\|$ decreases, which seems to be inconsistent with my argument above that also other points may be stationary points if they happen to have specific corresponding $h_t$ values.
> > > >
> > > > Can you please adapt the proof of Theorem 1 so it is consistent with your change in Line 8 of algo 2, and so that it is much clearer about what you are trying to achieve, ideally avoiding the $\infty$ or else explaining everywhere how this factors in ?

---

> > > > > ### Author Response · Authors · 2023-12-28
> > > > > **Authors' reply**
> > > > >
> > > > > Thank you for the detailed comments. We appreciate a very detailed discussion. Please see our response below.
> > > > >
> > > > > Regarding your first comment, your reasoning is correct. However, this can't happen due to the input condition (Line 1) that requires the sum of $h^{(i)}_0$'s is always zero. Furthermore, the update on Line 13 of Algorithm 2 also guarantees that the sum of $h^{(i)}_t$'s is always zero for any $t$. Thus, what the reviewer describes can only happen in the case $x^C = x^\star$, which is a desirable property.
> > > > >
> > > > > Regarding the proof:
> > > > > - Regarding intuition, our analysis mimics the analysis that one would usually use to analyze methods with variance reduction. We will add more comments, and we would appreciate it if the reviewer could point out places in our proof that need more explanation.
> > > > > - As we argued in our prior response, this is equivalent since averaging is in a linear operator and $h_t$ averages to zero.
> > > > > - To the best of our knowledge, we do not face any issue regarding infinity since the function is always evaluated on the same point for all the clients. Furthermore, we guarantee convergence in terms of distance to the solution which is always finite. Could you please point out where the issue in our analysis might arise?
> > > > > - Please note that the first argument does contain $\tilde{y}_t$ not $y_t$, which causes the difference.
> > > > >
> > > > > Regarding the last point, please see the first part of our response.
> > > > >
> > > > > We will incorporate all the requested changes into the camera-ready manuscript.

---

> > > > > > ### Comment · Reviewer_24ts · 2023-12-30
> > > > > >
> > > > > > I am aware that in the first iteration $\sum_i h_0^i=0$ as indicated in line 1.  Do you prove somewhere that $\sum_i h_t^i=0$ also for $t>1$ ?
> > > > > > * if yes, please specify where
> > > > > > * if bo, please specify exactly what you claim "can not happen"
> > > > > >
> > > > > >
> > > > > > As said previously, after your proposed change to line 8 the proof isnt consistent anymore with the new algorithm.  I pointed to a few first corrections but until i see a revised version it is hard to provide further comments.
> > > > > >
> > > > > > Concerning infinity, if you want to understand the problem, check when the function you define outputs infinity and then either argue in the proof why this will never happen or explain in the proof why the argument still holds if the function outputs infinity.
> > > > > >
> > > > > > I dont know if this tmlr allows to update the submission and or how long this discussion stays open.  Without revised version i can only say i think the currently available version is flawed.

---

> > > > > > > ### Author Response · Authors · 2024-01-07
> > > > > > > **Response**
> > > > > > >
> > > > > > > Dear Reviewer 24ts,
> > > > > > >
> > > > > > > Regarding your first point, we have submitted an updated version, where we have revised the update rule for $h^{(i)}_t$'s such that it is the case even when we use clipping, we always have $\frac{1}{N}\sum h^{(i)}_t = 0$ (for the previous version, this was only the case without clipping). The appendix contains detailed proofs, including a derivation of why it always holds $\frac{1}{N}\sum h^{(i)}_t = 0$ (which implies consistency with the previous version of the algorithm) and why the infinity value is not an issue since we never evaluate the reformulated problem.
> > > > > > >
> > > > > > >
> > > > > > > Please let us know if this addresses your concerns and whether there are any unanswered/unclear questions.
> > > > > > >
> > > > > > > Best,
> > > > > > > Author(s)

---

> > > > > > > > ### Comment · Reviewer_24ts · 2024-01-07
> > > > > > > >
> > > > > > > > Ok, I see now that in line 8 the update rule is changed as described in one of the above comments.
> > > > > > > > You now say that you also change the update rule for the $h_i^{(t)}$, I guess you mean you changed line 13.
> > > > > > > >
> > > > > > > > if in line 7 the "else" branch is chosen, it is well possible that not all $x_t^{(i)}$ are equal (for all i).  In that case, $\phi(x)=\infty$ will hold and we'll have the problems I referred to above.  I guess this can be solved by explicitly stating in the proof that you only consider the sub-sequence of $x_t$ for which $\theta_t=1$.  Nevertheless, in that case the probability $p$ in Eq (4) may be strange (as it should be 1 for a sub-sequence with $\theta_t=1$).  At least, if $\theta_t=0$, the minimization problem defined at the start of Sec B.1 will be strange, i.e., the objective function will be infinity for times $t$ where $\theta_t=0$ and you won't get a converging sequence of objective function values over time.
> > > > > > > >
> > > > > > > > Now I'm trying to figure out what is $T_1$ and $T_2$ in Eq (4).
> > > > > > > > I guess that with Eq (4) you want to say that with probability $(1-p)$, $\psi_{t+1}=\psi_t$, and with probability $p$ we have $\psi_{t+1}=\mathbb{E}[T_1+T_2]$ where $T_1$ and $T_2$ are defined in some way.
> > > > > > > >
> > > > > > > > I next guess that $T_1$ is meant to be $\|x_{t+1}-x^*\|^2$.  The proof here then suggests that $x_{t+1}=A(\hat{x}_{t+1} - (\eta/p) h_t + e_t) $
> > > > > > > >
> > > > > > > > However,  this is not consistent with line 9 of algorithm 2 which computes $x_{t+1}$ as
> > > > > > > > $x_t+\sum_{j=1}^N \Delta_t^{(j)}$
> > > > > > > > and does not let $x_{t+1}$ depend on ${\hat{x}}_{t+1}$.
> > > > > > > >
> > > > > > > > Can you please clarify the details around Eq (4)?

---

> > > > > > > > > ### Author Response · Authors · 2024-01-08
> > > > > > > > >
> > > > > > > > > Dear Reviewer 24ts,
> > > > > > > > >
> > > > > > > > > Thank you for your engaging discussion with us.
> > > > > > > > >
> > > > > > > > > Regarding your first comment, it is indeed the case that we have updated line 13 of Algorithm 2 (DP-ScaffNew), such that it is always the case that $\sum h^{(i)}_t = 0$ even when we use clipping.
> > > > > > > > >
> > > > > > > > > Regarding the infinity, we understand your reasoning. However, please note that we do not work with functional values of $\phi(x)$, but we provide convergence with respect to the distance of the current solution to the optimal solution, which is always finite. Thus, our analysis applies in every step. On the other hand, it is the case that we cannot guarantee convergence in terms of functional value since $\phi(x) = \infty$ for the case $\theta_t = 0$, as the reviewer argues. We would be only able to extend the analysis to the functional values of the steps, where $\theta_t = 1$ (i.e., averaging step).
> > > > > > > > >
> > > > > > > > > Regarding Eq. (4), please see below:
> > > > > > > > >
> > > > > > > > > Your reasoning is correct here. With probability $1-p$, $h_t$ stays the same, and
> > > > > > > > >
> > > > > > > > > \begin{align}
> > > > > > > > > \mathbf{x_{t+1}} \to \mathbf{\hat{x}_{t+1}},
> > > > > > > > > \end{align}
> > > > > > > > >
> > > > > > > > > and with probability $p$, we update the Lyapunov function due to averaging. $T_1$ corresponds to the first term of the Lyapunov function and $T_2$ to the second term. For $T_1$, we have updated paper with the derivation. Please see the first part of the proof, where we derive this equivalence.
> > > > > > > > >
> > > > > > > > > Best,
> > > > > > > > >
> > > > > > > > > Author(s)

---

> > > > > > > > > > ### Comment · Reviewer_24ts · 2024-01-08
> > > > > > > > > >
> > > > > > > > > > The problem remains that the proof does not mention once the clipping function, and hence does not provide a relation between the variables equivalent to Algorithm 2.
> > > > > > > > > >
> > > > > > > > > > For example, the text says:
> > > > > > > > > >
> > > > > > > > > > > Algorithm 2 can be expressed equivalently as: $x_{t+1} = A(\hat{x}_{t+1}+e_t )$ with probability p.
> > > > > > > > > >
> > > > > > > > > > This is not correct, as in reality Algorithm 2 does not average ${{\hat x}_{t+1}}+e_t$,
> > > > > > > > > >
> > > > > > > > > >  but averages $x_t$ plus the *clipping* of $\hat{x}_{t+1}-x_t$ (plus $e_t$).
> > > > > > > > > >
> > > > > > > > > > As long as the proof is not consistent with the Algorithm, it is hard to believe it correctly proves a property of the algorithm.

---

> > > > > > > > > > > ### Author Response · Authors · 2024-01-08
> > > > > > > > > > >
> > > > > > > > > > > Dear Reviewer,
> > > > > > > > > > >
> > > > > > > > > > > This is the case and limitation of our analysis, not something we are trying to hide. This is discussed in the last paragraph of page 4. We mention several times that we provide the analysis assuming clipping is never active, which is where our analysis holds. The clipping operator introduces significant non-linearity into the updates, greatly complicating the analysis. However, our experimental results demonstrate that as the algorithm converges, the magnitudes of the updates diminish, and clipping primarily impacts only the initial rounds. Consequently, in practical scenarios, running the algorithm with or without clipping has a minimal effect on convergence; see Figure 11 in the appendix.
> > > > > > > > > > >
> > > > > > > > > > > We would also like to highlight that this is a common issue in DP literature, e.g., see a recent work of Choquette-Choo et al., 2023 (https://arxiv.org/pdf/2310.06771.pdf). Therefore, we would like to request the reviewer to reconsider their evaluation and not consider this limitation fatal. It would be interesting to have an analysis where we can argue also about the effect of clipping, but this is currently out of the scope of our work.
> > > > > > > > > > >
> > > > > > > > > > > Best,
> > > > > > > > > > >
> > > > > > > > > > > Author(s)

---

> > > > > > > > > > > > ### Comment · Reviewer_24ts · 2024-01-08
> > > > > > > > > > > >
> > > > > > > > > > > > It is true that in the first rounds most clipping is needed and this decreases a bit as time evolves.
> > > > > > > > > > > > Still, clipping does not fully disappear as time progresses.
> > > > > > > > > > > >
> > > > > > > > > > > > It is possible that on the datasets you selected you observe (as you say) clipping primarily only impacts the initial rounds.  However, this is not what we observe in our experiments for all our datasets.  Maybe you were lucky in your selection of datasets and this observation holds less generally than you believe. (in fact, the number of datasets in the paper is not so large we can expect to easily generalize to other datasets)
> > > > > > > > > > > >
> > > > > > > > > > > > More fundamentally, even if somewhere you state you assume that clipping does not happen, clipping is used in Algorithm 2 and Theorem 1 formally claims a property of Algorithm 2 (without stating any conditions about clipping not happening), which as you admit is not true because it only holds in the cases where the assumption "clipping does not happen" holds.
> > > > > > > > > > > >
> > > > > > > > > > > > It may even turn out to be difficult to prove theorem 1 if you would add the condition "no clipping happens after round 15", since the excess gradients are accumulated in the $h_t$ and unless you can bound the gradient minus its clipped version you have no upper bound on the number of iterations it takes before the $h_t$ buffer is emptied again.
> > > > > > > > > > > >
> > > > > > > > > > > > I admit I have read Theorem 1 as it is claimed in the paper, not taking into considerations some extra conditions some other paragraphs may mention without making them formal in the theorem statement.  Making this (fourth!) substantial change to your paper since submission again seriously changes your claim.  Now the paper has no theoretical contribution anymore, as the theorem is rather trivial is you really assume that clipping never happens (not even in the first 15 rounds, as I argue above).
> > > > > > > > > > > >
> > > > > > > > > > > > The paper still has several disturbing unclarities as the ones identified above, and they would need to be fixed all if the paper would get accepted, but I fundamentally believe Theorem 1 under the assumption clipping never happens (not even in the first round), and so with sufficient work of the authors (it is not my task to edit the paper) these problems are likely fixable.
> > > > > > > > > > > >
> > > > > > > > > > > > However, the paper now becomes entirely an empirical result.    A new algorithm is proposed, the authors need to provide the empirical evidence that in practice it works better.  For this, a very in-depth empirical evaluation would be needed, using a sufficient wide range of datasets and tackling the experimental setup in a much more systematic way.
> > > > > > > > > > > >
> > > > > > > > > > > > * Observations 1 and 2 are not really surprising, many hyperparameters have an optimal value and for the optimal number of steps it is intuitively the case.
> > > > > > > > > > > > * Observation 3 is interesting and deserves more investigation
> > > > > > > > > > > > * Observation 4 shows an important flaw.  We just assumed clipping does not happen (or at least does not have a significant influence), so we should not observe that the clipping threshold has an influence.  (of course, you could also drop Theorem 1 or improve the formulation of Theorem 1 or otherwise make things consistent)
> > > > > > > > > > > >
> > > > > > > > > > > > In conclusion, while I believe it may be possible to do something to decrease the negative impact of the clipping the current paper does not seem to touch this interesting question (despite the current formulation of Theorem 1 giving that impression).  Rereading Section 1 I'm trying to understand what now actually the contribution is, but I don't see a contribution which is supported by extensive experimental evidence.

---

> > > > > > > > > > > > > ### Author Response · Authors · 2024-01-09
> > > > > > > > > > > > >
> > > > > > > > > > > > > Dear Reviewer 24ts,
> > > > > > > > > > > > >
> > > > > > > > > > > > > We appreciate your feedback and would like to address your concerns succinctly.
> > > > > > > > > > > > >
> > > > > > > > > > > > > - **Regarding Theoretical Contributions:** We maintain that our paper contributes significantly to theoretical understanding. Theorem 1 explicitly states all assumptions, including those about clipping (refer to line 3 of the theorem). Our theoretical framework, albeit simplified, provides meaningful insights consistent with standard practices in ML research. Simplifications, like assuming convexity in neural networks, are common to facilitate understanding of complex systems. Our assumptions serve a similar purpose.
> > > > > > > > > > > > >
> > > > > > > > > > > > > - **On the Consistency of Our Theory with Clipping:** Our theoretical model, while simplified, remains applicable even if clipping is active during a finite number of initial rounds. Due to the smoothness of the algorithm, it cannot diverge within these rounds. Therefore, our analysis can consider the starting point as the one where the last clipping was applied, which is finitely distanced from the initial point.
> > > > > > > > > > > > >
> > > > > > > > > > > > > - **On Changes Concerning Clipping:** The modifications made to our paper pertain only to the aspect of clipping. It is crucial to note that the algorithm operates identically to its initial version without clipping. Furthermore, for overparametrized models (the main focus of work), none of the changes make a difference as we do not need to work with $h_t^{i}$ as we have $h_t^{i} = \nabla f_i(x^*) = 0$ and DP-ScaffNew becomes DP-FedAvg.
> > > > > > > > > > > > >
> > > > > > > > > > > > > We hope these clarifications address your concerns and reinforce the value of our paper. Our work offers insights into the algorithm's theoretical and empirical behavior under various conditions.
> > > > > > > > > > > > >
> > > > > > > > > > > > > We respectfully request a reconsideration of the claim that our paper lacks theoretical contribution.
> > > > > > > > > > > > >
> > > > > > > > > > > > > Best,
> > > > > > > > > > > > >
> > > > > > > > > > > > > Author(s)

---

> > > > > > > > > > > > > > ### Comment · Reviewer_24ts · 2024-01-09
> > > > > > > > > > > > > >
> > > > > > > > > > > > > > It is true that after a finite number of initial clipping rounds, the distance of $x_t$ to the starting point and to the optimal point will be finite, but there is no bound on it as there is no bound on the gradient.  As such, there is no known bound on the term $\theta^\top \psi_0$ on the righthandside of Eq (2), and the theorem isn't very useful anymore with an unbounded righthandside.
> > > > > > > > > > > > > >
> > > > > > > > > > > > > > You claim that you make theoretical contributions in the case where there is no clipping, but as far as I understand the setting where there is no clipping is already investigated in quite some depth.  On page 2 you list the contribution "we analyze the DP version of ..." but in fact you just mean "we analyze the noisy version of ...".
> > > > > > > > > > > > > >
> > > > > > > > > > > > > > While it is a standard practice to consider convex functions for the ease of analysis, assuming clipping never happens is an assumption which is much less frequently made.  In such cases, usually authors are more transparent and just say they study a noisy version without introducing and immediately discarding clipping.  In fact, an additional problem is that it is hard to predict from a given dataset whether clipping will ever occur.   For existing algorithms often a data owner can evaluate for an instance whether the gradient will trigger clipping, and this could allow one to choose a good value for $C$ from the start.  In Algorithm 2 however, we don't clip the gradient but the gradient augmented with $h_t^{(i)}$.  So depending on the evolution of the parameters $\theta$ this $h_t$ may accumulate so much gradient that at some later point clipping occurs, and we don't have an easy way to check at the start whether this will happen.
> > > > > > > > > > > > > >
> > > > > > > > > > > > > > In fact, while looking at a bound for $h_t$, I notice that Algorithm 2 has still a problem.  If we get in the "else" branch and in line 11 we skip communication, how then in line 13 the clients are able to update $h_{t+1}^{(i)}$ using the average $\Delta_t^{(j)}$ over all clients j?  "Skipping communication" suggests the clients are not supposed to have that information.

---

> > > > > > > > > > > > > > > ### Author Response · Authors · 2024-01-11
> > > > > > > > > > > > > > >
> > > > > > > > > > > > > > > Thank you for the detailed comments!
> > > > > > > > > > > > > > >
> > > > > > > > > > > > > > > Regarding the first point, assuming the smoothness, the gradient must also be finite with a finite number of steps.
> > > > > > > > > > > > > > >
> > > > > > > > > > > > > > > Regarding the point about analysis without clipping. To the best of our knowledge, our analysis is the first one that shows that the DP version without clipping (i.e., noisy) of the FedAvg-like algorithm can benefit from local steps similar to the plain ScaffNew algorithm. On top of that, our theory reveals that the optimal number of local steps is independent of the noise level $\sigma^2$. We are not aware of any other work with similar insights. Therefore, we kindly disagree with the claim that this is already well-studied.
> > > > > > > > > > > > > > >
> > > > > > > > > > > > > > > Regarding your point about $h^{(i)}_t$'s causing a problem. Please note that $h^{(i)}_t$'s act as a variance reduction (which we prove theoretically). Thus, it is natural to expect that in practice, if we do not clip due to gradient updates, we would not clip when including $h^{(i)}_t$'s as these act as variance reduction, which is also consistent with what we observed in the experiments. Also, we could rewrite all our theoretical claims as applied to noisy ScaffNew for the camera-ready version instead of DP if the reviewer requires this. However, please note that all the assumptions are several times mentioned in the main text and reiterated in the theorem formulation.
> > > > > > > > > > > > > > >
> > > > > > > > > > > > > > > For the last point, thank you for spotting this typo caused by updating Algorithm 2 with $\Delta$'s instead of iterates differences for better readability. When there is no communication, $h^{(i)}_t$'s are not updated, which is how we implement and analyze Alg. 2 already (please recall our answer to your question about $T_1$ and $T_2$). This is now fixed in the Algorithm 2.

---

### Review · Reviewer_YJVi · 2023-11-10

**Summary Of Contributions:**

The paper scrutinizes the delicate equilibrium between preserving data privacy and maintaining high performance in federated learning systems. It makes the case that there exists a specific, optimal combination of the number of local updates performed by client devices and the frequency of communication between these devices and the central server, which can significantly mitigate the performance loss typically associated with implementing differential privacy measures. This optimal balance ensures that the performance of the federated learning model remains robust while still adhering to stringent privacy constraints. The authors substantiate their claims with a thorough analysis and empirical evidence, showing that with careful tuning, differentially private federated learning algorithms can function almost as effectively as their non-private counterparts.

**Audience:**

Yes

**Claims And Evidence:**

Yes

**Requested Changes:**

(1). How does the complexity of tuning the hyper-parameters impact the practical deployment of your differentially private federated learning algorithms in real-world scenarios?
(2). Your results focus on strongly convex problems. How well do you expect the proposed optimal hyper-parameters to perform on non-convex optimization problems, which are prevalent in many machine learning tasks?
(3). How sensitive are the optimal hyper-parameters to changes in the privacy budget, and what would be the process for adjusting these parameters if the privacy requirements change?

**Strengths And Weaknesses:**

Strengths:
	The paper successfully identifies an optimal set of hyper-parameters for DP federated algorithms that balance privacy and performance.
	It provides a thorough analysis of the DP version of the ScaffNew algorithm under non-restrictive assumptions and finds explicit expressions for the optimal number of total communication rounds.
	The findings are applicable to strongly convex optimization without the need for data heterogeneity assumptions, broadening the scope of the algorithm’s application.

Weaknesses:
	Complexity in Hyper-parameter tuning. The need for careful tuning of the hyper-parameters might introduce complexity in practical applications where such precision is difficult to achieve.
	The integration of differential privacy mechanisms could add computational overhead, potentially making the algorithms less efficient in terms of computation time and resource usage.
	Even though it is minimized, there is still a trade-off between privacy and performance, which might not be acceptable in all use cases.

---

> ### Author Response · Authors · 2023-12-15
> **Author's Response**
>
> We would like to thank the reviewer for their valuable feedback and time spent reviewing our manuscript. We appreciate that the reviewers found many strong points in our manuscript, including:
>
> - Identification of an optimal set of hyper-parameters for DP federated algorithms that balance privacy and performance.
> - A thorough analysis of the DP version of the ScaffNew algorithm under non-restrictive assumptions.
> - The findings are applicable to strongly convex optimization without the need for data heterogeneity assumptions, broadening the scope of the algorithm’s application.
>
> We address all the reviewer's concerns below:
>
> >  How does the complexity of tuning the hyper-parameters impact the practical deployment of your differentially private federated learning algorithms in real-world scenarios?
>
> Thank you for this important comment. Our theory reveals there is a non-trivial number of local steps and total communication rounds for which the DP methods achieve the best performance. Furthermore, it is not only the case that these values exist, but we also give the formula for how to compute them. On top of that, we note that it is hard to do any tuning for DP methods as one needs to use much smaller $\epsilon$ in order to guarantee the same DP bound while running multiple rounds of tuning hyperparameters. However, our optimal formulas offer a novel solution to this issue. We note that the optimal number of local steps is independent of $\epsilon$, and for the optimal number of communication rounds, we only have a factor $\ln{\epsilon}$ that needs to be adjusted. Therefore, our results suggest that one can tune hyperparameters with much higher privacy guarantees, while we only run with target privacy guarantee once, already with tuned parameters. We believe that such an implication is an important contribution that is also well-aligned with our experimental evaluation.
>
>
> > Your results focus on strongly convex problems. How well do you expect the proposed optimal hyper-parameters to perform on non-convex optimization problems, which are prevalent in many machine learning tasks?
>
> Thank you for raising this important question!
> While our theory holds under the strong convexity assumption, we have extended our empirical evaluation to include both convex and non-convex problems such as FEMNIST, Reddit, and CIFAR10. On top of that, we also evaluated our approach on real-world datasets, including diabetic retinopathy grade classification and MRI brain segmentation, to assess the practical utility of our approach.
> Our results indicate that the theoretical insights underlying our proposed DP-Scaffnew are not confined to convex scenarios. They show promising applicability and performance even in non-convex models. The consistency we observed in balancing privacy preservation with the number of local updates in these diverse contexts underscores the versatility of our method.
>
> > How sensitive are the optimal hyper-parameters to changes in the privacy budget, and what would be the process for adjusting these parameters if the privacy requirements change?
>
>  Please see our response to your first question.

---

### Review · Reviewer_6cRV · 2023-12-01

**Summary Of Contributions:**

The paper presents a Differentially Private version of the ScaffNew algorithm. It analyzes the utility guarantee of DP-ScaffNew for a strongly convex case with the assumption that (a) each local step is based on the full gradient and (b) the clipping operator is never active. Further, the optimal values of $\eta, p, T$ are derived for the above case.

The authors presented empirical results of DP-FedAvg and DP-ScaffNew on 5 different datasets with the total number of clients varying from 3-6. From these results, the authors make four observations:
1. There exists an optimal number of local steps (defined by p) for DP-FedAvg and DP-ScaffNew.
2. There exists an optimal number of total iteration steps (T) for DP-FedAvg and DP-ScaffNew.
3. The optimal number of local steps increases and the optimal total iterations number decreases as $\epsilon$ decreases.
4. The optimal local steps depend on the clipping threshold, but it does not significantly impact performance.

**Audience:**

Yes

**Claims And Evidence:**

No

**Requested Changes:**

1. Add the details on how Theorem 1 from [1] is extended to Lemma 1 presented in the paper.
2. Consider adding the utility guarantees for a convex case with standard assumptions similar to [2].
3. Consider conducting experiments on larger client sizes.
4. Conduct comparisons with recent methods such as DP-Scaffold.

[1] Abadi, Martin, et al. "Deep learning with differential privacy." Proceedings of the 2016 ACM SIGSAC conference on computer and communications security. 2016.

[2] Noble, Maxence, Aurélien Bellet, and Aymeric Dieuleveut. "Differentially private federated learning on heterogeneous data." International Conference on Artificial Intelligence and Statistics. PMLR, 2022.

**Strengths And Weaknesses:**

Strengths:
1. Theoretically analyzed the convergence of DP-ScaffNew for strongly convex case.
2. The paper provides the explicit expression for optimal hyper-parameters for DP-ScaffNew $\eta^⋆, p^⋆, T^⋆$ for strongly convex case.
3. Empirical results of DP-FedAvg and DP-ScaffNew on 5 different datasets and various model architectures are presented.
4. Empirical evaluation shows the correlation between the optimal number of iterations and local steps with the privacy budget.

Weaknesses/questions:
1. One would expect that setting $C=\infty$ and $\sigma=0$ in DP-ScaffNew recovers the classical ScaffNew algorithm. But this is not the case with Algorithm 2. The global averaging step of DP-ScaffNew becomes $x_{t+1} = \frac{1}{N} \sum_{i=1}^{N}(\hat{x}^i_{t+1} - \frac{\eta}{p}h^i_t)$ when we set $C=\infty$ and $\sigma=0$. However, ScaffNew's update rule is $x_{t+1} = \frac{1}{N} \sum_{i=1}^{N}\hat{x}^i_{t+1} $. What is the significance of the additional $\frac{\eta}{p}h^i_t$ introduced in DP-ScaffNew.
2. In lemma 1, why is the sampling probability ($q$ from Theorem 1 of [1]) replaced by $C^2 p$? It is not clear to me why the clipping ratio $C$ is included in the bound.
3. The theory for the strongly convex case is presented under the assumption that each local step is based on the full gradient. This is not a standard assumption and is not practical.
4. The optimal values for p and T are derived for the strongly convex case but there is no explanation on how to interpret these theoretical results. In particular, what did we learn from these optimal values?
5. observations 1 and 2 mention that there exists a non-trivial optimal value for the number of local steps and iterations. Can authors clarify what exactly contributes to the non-trivial nature of these observations?
6. From Figure 3, it is not clear that the number of iterations decreases with an increase in privacy degree. The authors mention that observation 3 regarding local steps doesn't match with theoretical results. Can authors elaborate on the reasons?
7. Why are the experiments conducted for a very small number of clients? Is the proposed DP-ScaffNew method not scalable? Do the observations from section 3 still hold if we increase the number of clients?
8. What value of $p$ is used in the experiments?

[1] Abadi, Martin, et al. "Deep learning with differential privacy." Proceedings of the 2016 ACM SIGSAC conference on computer and communications security. 2016.

[2] Noble, Maxence, Aurélien Bellet, and Aymeric Dieuleveut. "Differentially private federated learning on heterogeneous data." International Conference on Artificial Intelligence and Statistics. PMLR, 2022.

---

> ### Author Response · Authors · 2023-12-15
> **Author's response**
>
> We would like to thank the reviewer for their valuable feedback and time spent reviewing our manuscript. We appreciate that the reviewers found many strong points in our manuscript, including:
>
> - An optimal number of local steps and iterations exists for DP-FedAvg and DP-ScaffNew.
> - The paper provides the explicit expression for these optimal hyper-parameters.
> - Empirical results of DP-FedAvg and DP-ScaffNew on 5 different datasets and various model architectures are presented.
> - Empirical evaluation shows the correlation between the optimal number of iterations and local steps with the privacy budget.
>
> We address all the reviewer's concerns below:
>
> > ScaffNew as a special case of DP-ScaffNew.
>
> This is equivalent to ScaffNew as, by construction, it is always the case that the sum of $ h^i_t$’s is zero.
>
> > In lemma 1, why is the sampling probability ($q$ from Theorem 1 of [1]) replaced by $pC^2.$ It is not clear to me why the clipping ratio
>  is included in the bound.
>
> In Lemma 1, we use $q=1$ (the worst case of Theorem 1 of [1]), and $p$ belongs to $T$ as the total number of communication rounds is $pT$, not $T$. Regarding the multiplication by the gradient clipping threshold $C$, this difference stems from our algorithm, where we do not explicitly multiply by the gradient clipping threshold $C,$ as was done in the original theorem. This variation is reflected in our formulation.
>
> [1] Abadi, Martin, et al. "Deep learning with differential privacy." Proceedings of the 2016 ACM SIGSAC conference on computer and communications security. 2016.
>
> > The theory for the strongly convex case is presented under the assumption that (a) each local step is based on the full gradient and (b) the clipping operator is never active. These are not the standard assumptions and are not practical.
>
> As discussed in the manuscript, assumption (A) is not considered essential when we assume overparametrization of the models, meaning they can perfectly fit the data. However, in cases where overparametrization is not present, we acknowledge that this assumption may be impractical. The development of an extension of our proposed algorithm, which neither requires full gradient computation nor relies on overparametrization, is a promising direction for future research and merits a separate paper. It's important to note that, to the best of our knowledge, even the ScaffNew algorithm without differential privacy (DP) lacks guarantees for local stochastic gradient computation.
>
> Regarding the second assumption (B), its importance cannot be overstated. The clipping operator introduces significant non-linearity into the updates, greatly complicating the analysis. However, our experimental results demonstrate that as the algorithm converges, the magnitudes of the updates diminish, and clipping primarily impacts only the initial rounds. Consequently, in practical scenarios, running the algorithm with or without clipping has a minimal effect on convergence.
>
> In conclusion, our proposed theory offers new theoretical insights despite the necessity for stronger assumptions, which align with those in prior work. Exploring ways to relax these assumptions is an intriguing area for future research but is beyond the scope of our current work.

---

> ### Author Response · Authors · 2023-12-15
> **Authors' Response Contd. 2**
>
> > Why are the experiments conducted for a very small number of clients? Is the proposed DP-ScaffNew method not scalable? Do the observations from section 3 still hold if we increase the number of clients?
>
> Thank you for your question regarding the number of clients in our experiments. Our study primarily focuses on cross-silo federated learning scenarios, where the number of clients is typically more limited compared to cross-device settings. Additionally, in these contexts, each client often possesses a larger dataset. However, it is  important to note that both our theoretical framework and experimental validations are not dependent on the number of clients. Therefore, we do not expect the issues with scalability of our proposed DP-ScaffNew method. It would be interesting to further extend our work to cross-device FL. However, this would first require non-trivial theoretical extension to include partial participation into the DP-ScaffNew algorithm. We plan to investigate this as a future work, but it is currently out of the scope of our current study.
>
> > What value of p is used in the experiments?
>
> Thank you for your question regarding the value of $p$ in our experiments. In our experiments, we decided to use the notion of local steps rather than $p$ as this is more common in FL. For our experiments, $p = 1 / \textit{the number of local steps}$, as the expected number of local steps is $1/p$. We hope that this clarifies your concern.
>
> > $\sqrt{\log{T}}$ vs. $\sqrt{T}.$
>
> We kindly disagree with the reviewer’s claim. As the authors of [2] clarify straight after stating Assumption 1, which refers to $ \sqrt{\log{T}}$ pointed by the reviewer:
>
>
> "Note that our analysis **does not require** Assumption 1, but the resulting expressions and the dependency on the key parameters are then difficult to interpret. This assumption is actually **not used in our experiments,** where we compute the privacy loss numerically **using the complete formulas** from our proof."
>
> where the **complete formulas** refer to Theorem 4.1 in their paper, where noise scales as $\sqrt{T \times \log{T}}$ using advanced accounting, thus the reviewer’s claim about $ \sqrt{\log{T}}$ is not correct and at least $\sqrt{T}$ is required. Finally, please not that our theory is somehow independent of accounting and if there was a better accounting method, we could directly apply it on top of our DP-ScafNew to obtain better dependence of $T$ for the noise $\sigma$.
>
> > Add the details on how Theorem 1 from [1] is extended to Lemma 1 presented in the paper.
>
> Please see our response to your concern 2.
>
> > Consider adding the utility guarantees for a convex case with standard assumptions similar to [2].
>
> Convex case is currently out of the scope of our current work. We are more interested in extending our current theory into Polyak-Łojasiewicz settings that can also capture structured non-convexity.
>
> > Consider conducting experiments on larger client sizes.
>
> As mentioned by the reviewer, we already conducted experiments on a wide variety of tasks (5). As we discussed, before extending to cross-device FL, we would first require a non-trivial theoretical extension to include partial participation in the DP-ScaffNew algorithm that we leave for future work.
>
> > Conduct comparisons with recent methods such as DP-Scaffold.
>
> DP-ScafNew is algorithmically almost equivalent to DP-Scaffold. Therefore, it is not meaningful to compare these two in the experiments as this would lead to exactly the same results. However, our theory improves upon DP-Scaffold as we are able to provide the exact formulas for the optimal number of local steps and iterations. On top of that, we can show that DP-ScafNew directly benefits from local steps, i.e., $p=1$ is not optimal, which is not the case for DP-Scaffold unless the similarity across the clients is assumed, where our theory allows for arbitrary heterogeneity across clients. Note that the optimal local step size scales as $\mathcal{O}(1/\textit{the number of local iterations})$ for DP-Scaffold, while for DP-ScafNew, it scales as $\mathcal{O}(1)$.

---

### Decision · Action_Editor_wYGh · 2024-01-19

**Recommendation:** Reject

**Comment:**

My primary concern with accepting the paper is the discussion between Reviewer 24ts and the authors. As I wrote above, the omission of clipping means that the paper fundamentally does not study DP federated algorithms. Note that while there are other works on algorithms for differential privacy that omit clipping, the paper specifically brought up in the author/reviewer discussion (https://arxiv.org/pdf/2310.06771.pdf) is about comparing different patterns of noise added by various DP methods. By contrast, the work being reviewed is explicitly about "optimal number[s] of local steps" for "the DP version of the ScaffNew algorithm." While this distinction is subtle, I think it is important.

The discussion with Reviewer 24ts also revealed discrepancies between Algorithm 2 (as stated) and the algorithm analyzed in the theoretical results. In general, the authors have proposed multiple revisions of specific lines of the paper that when taken in sum, I believe point to the need for a significant revision to the paper, especially as it is unclear (based on reviewer discussion) whether Theorem 1 is correct. The revised version of the work that purports to show that the average of the control variates $h_t^{(i)}$ are 0 at every round $t$, but was not included in the original submission and would require re-vetting the theory to a significant degree.

As stated above, without the theoretical results, the empirical results are likely of limited interest. Therefore, I do not recommend acceptance.

**Audience:**

While there are certainly TMLR audiences who would be interested in papers that show that "DP federated algorithms have the optimal number of local steps and communication rounds to balance privacy and convergence performance," I do not believe this has been shown sufficiently clearly that this is a salient factor in acceptance of the work. Additionally, as Reviewers 24ts and 6cRV point out, without the theory the empirical results are of limited interest due to their relatively limited scope.

**Claims And Evidence:**

The paper is not currently at a state where I can say that all the claims made in the paper are backed by clear evidence. In particular, the discussion between Reviewer 24ts and the authors was illuminating about clipping and differential privacy. The authors state in the paper that "we prove the optimal number of local steps and communication rounds that enhance the convergence bounds of the DP version of the ScaffNew algorithm". However, this critically relies on an assumption that "the clipping operator is never active". While there was some discussion about how reasonable this assumption is (potentially bolstered by the empirical results), I do not believe that the authors have clearly justified that with this assumption, their theory is relevant to actual differential privacy algorithms.

**Resubmission Of Major Revision:**

The authors may consider submitting a major revision at a later time.